# GAIN: ENHANCING BYZANTINE ROBUSTNESS IN FEDERATED LEARNING WITH GRADIENT DECOMPOSITION

## ABSTRACT

Federated learning provides a privacy-aware learning framework by enabling participants to jointly train models without exposing their private data. However, federated learning has exhibited vulnerabilities to Byzantine attacks, where the adversary aims to destroy the convergence and performance of the global model. Meanwhile, we observe that most existing robust AGgregation Rules (AGRs) fail to stop the aggregated gradient deviating from the optimal gradient (the average of honest gradients) in the non-IID setting. We attribute the reason of the failure of these AGRs to two newly proposed concepts: identification failure and integrity failure. The identification failure mainly comes from the exacerbated curse of dimensionality in the non-IID setting. The integrity failure is a combined result of conservative filtering strategy and gradient heterogeneity. In order to address both failures, we propose GAIN, a gradient decomposition scheme that can help adapt existing robust algorithms to heterogeneous datasets. We also provide convergence analysis for integrating existing robust AGRs into GAIN. Experiments on various real-world datasets verify the efficacy of our proposed GAIN.

## 1 INTRODUCTION

Federated Learning (FL) (McMahan et al., 2017) is a privacy-aware distributed machine learning paradigm. It has recently attracted widespread attention as a result of emerging data silos and growing privacy awareness. In this paradigm, data owners (clients) repeatedly use their private data to compute local gradients and send them to a central server for aggregation. In this way, clients can collaborate to train a model without exposing their private data. However, the distributed property of FL also makes it vulnerable to Byzantine attacks (Blanchard et al., 2017; Guerraoui et al., 2018). During the training phase, Byzantine clients can send arbitrary messages to the central server to bias the global model. Moreover, it is challenging for the central server to identify the Byzantine clients, since the server can neither access clients' training data nor monitor local training processes.

In order to defend against Byzantine attacks, the community has proposed a wealth of defenses (Blanchard et al., 2017; Guerraoui et al., 2018; Yin et al., 2018). Most defenses abandon the averaging step adopted by conventional FL frameworks, e.g., FedAvg (McMahan et al., 2017). Instead, they use robust AGgregation Rules (AGRs) to aggregate local gradients in order to defend against Byzantine attacks. Most existing robust AGRs assume that the data distribution on different clients is identically and independently distributed (IID) (Bernstein et al., 2018; Ghosh et al., 2019). However, the data is usually non-independent and identically distributed (non-IID) in real-world FL applications (McMahan et al., 2017; Karimireddy et al., 2020; Kairouz et al., 2021). As a result, in more realistic non-IID settings, most robust AGRs fail to defend against Byzantine attacks, and thus suffer from significant performance degradation (Karimireddy et al., 2022; Acharya et al., 2022).

To investigate the cause of the degradation, we perform a thorough experimental study on various robust AGRs. Close inspection reveals that the reason behind the degradation is different for different AGRs with different types of aggregation strategies. For *conservative* AGRs that only aggregate few gradients to get rid of Byzantines, they suffer from *integrity* failure. The integrity failure describes that an AGR can only identify *few* honest gradients for aggregation. This failure will lead to an aggregated gradient with limited utility due to the gradient heterogeneity (Li et al., 2020; Karimireddy

et al., 2020) in the non-IID setting. For *radical* AGRs that aggregate as many gradients as possible to avoid such deviation, they suffer from another *identification* failure. The identification failure means that an AGR fails to distinguish between honest and Byzantine gradients. This failure is mainly due to the curse of dimensionality (Guerraoui et al., 2018; Diakonikolas et al., 2017) aggravated by the non-IIDness. Both failures deviate the aggregated gradient from the optimal gradient (the average of honest gradients). As a result, most existing AGRs fail to achieve a satisfactory performance in the non-IID setting.

Motivated by the above observations, we propose a GrAdient decomposItioN method called GAIN that can handle both failures in various non-IID settings. In particular, to address the identification failure due to the curse of dimensionality, GAIN decomposes each high-dimensional gradient into low-dimensional groups for gradient identification. Then, GAIN incorporates gradients with low identification scores into final aggregation to tackle the integrity failure.

Our contributions in this work are summarized below.

- We reveal the root reasons for the performance degradation of current robust AGRs in the non-IID setting by proposing two new concepts: integrity failure and identification failure. Integrity failure origins from the gradient heterogeneity, and identification failure is a result of the aggravated curse of dimensionality in the non-IID setting.

- We propose a novel and compatible approach called GAIN, which applies robust AGRs on the decomposed gradients, followed by identification before aggregation, rather than directly operating on the original gradients as the existing defenses (Multi-Krum (Blanchard et al., 2017), Bulyan (Guerraoui et al., 2018), etc) do.

- We also provide convergence analysis for integrating existing robust AGRs into GAIN. In particular, we provide an upper bound for the sum of gradient norms.

- We also offer empirical experiments on three real-world datasets across various settings to validate the effectiveness and superiority of our GAIN.

## 2 RELATED WORKS

Byzantine robust learning is first introduced by Blanchard et al. (2017). Subsequently, a range of works study the robustness against Byzantine attacks by proposing various robust AGgregation Rules (AGRs) under the IID setting. Generally, we can classify the current robust AGRs into two categories: **conservative** AGRs and **radical** AGRs.

Typical conservative AGRs, including Bulyan (Guerraoui et al., 2018), Median (Yin et al., 2018), Trimmed Mean (Yin et al., 2018), etc., only aggregate few gradients to reduce the risk of the introduced Byzantine gradients. Bulyan (Guerraoui et al., 2018) applies a variant of trimmed mean as a post-processing method to handle the curse of dimensionality. Yin et al. (2018) theoretically analyze the statistical optimality of Median and Trimmed Mean. The radical AGRs, e.g., Multi-Krum (Blanchard et al., 2017), DnC (Shejwalkar & Houmansadr, 2021), incorporate as many gradients as possible to avoid such deviation. Multi-Krum is a distance-based AGR proposed by Blanchard et al. (2017). (Pillutla et al., 2019) discuss the Byzantine robustness of Geometric Median and propose a computationally efficient approximation. Shejwalkar & Houmansadr (2021) propose to perform dimensionality reduction using random sampling, followed by spectral-based outlier removal. Recently, a quantity of works (Allen-Zhu et al., 2020; Karimireddy et al., 2021; Farhadkhani et al., 2022) discuss the effect of distributed momentum to Byzantine robustness from different perspectives. However, in more realistic FL applications where the data is non-IID, the efficacy of these defenses are quite limited. They fail to obtain high-quality aggregated gradient in the non-IID setting, thus suffer from significant performance degradation.

Recently works have also explored defenses that can be applicable to the non-IID setting. Park et al. (2021) can only achieve Byzantine robustness when the server has a validation set, which compromises the privacy principle of the FL (McMahan et al., 2017). Data & Diggavi (2021) adapt a robust mean estimation algorithm to FL to combat Byzantines in the non-IID setting. However, it requires $\Omega(d^2)$ time ($d$ is the number of model parameters), which is unacceptable due to the high dimensionality of model parameters. El-Mhamdi et al. (2021) consider Byzantine robustness in the asynchronous communication and unconstrained topologies settings. Acharya et al. (2022) propose to

apply geometric median only to the sparsified gradients to save computation cost. Karimireddy et al. (2022) perform a bucketing process before aggregation to reduce the gradient heterogeneity. These methods guarantee convergence of SGD in the existence of Byzantines. However, convergence is not enough in the context of non-convex, high dimensional case of neural networks (Guerraoui et al., 2018). These methods lack of the guarantee that the aggregated gradient does not deviate from the optimal gradient (the average of honest gradients). As a result, they may lead to convergence towards ineffectual models.

## 3 NOTATIONS AND PRELIMINARIES

**Notations.** For any positive integer $n \in \mathbb{N}^+$, we denote the set $\{1, \ldots, n\}$ by $[n]$. The cardinality of a set $\mathcal{S}$ is denoted by $\#\mathcal{S}$ or $|\mathcal{S}|$. For a real number $x \in \mathbb{R}$, we use $|x|$ to denote the absolute value of number $x$. We denote the $\ell_2$ norm of vector $\boldsymbol{x}$ by $\|\boldsymbol{x}\|$. We use $[\boldsymbol{x}]_j$ to represent the $j$-th component of vector $\boldsymbol{x}$. The sub-vector of vector $\boldsymbol{x}$ indexed by index set $\mathcal{J}$ is denoted by $[\boldsymbol{x}]_{\mathcal{J}} = ([\boldsymbol{x}]_{j_1}, \ldots, [\boldsymbol{x}]_{j_k})$, where $\mathcal{J} = \{j_1, \ldots, j_k\}$, and $k = |\mathcal{J}|$ is the number of indices. For a random variable $X$, we use $\mathbb{E}[X]$ and $\mathrm{Var}[X]$ to denote the expectation and variance of $X$, respectively.

**Federated learning.** We consider the federated learning system with a center server and $n$ clients following Blanchard et al. (2017); Yin et al. (2018); Guerraoui et al. (2018). Then the objective is to minimize loss $\mathcal{L}(\boldsymbol{w})$ defined as follows.

$$\mathcal{L}(\boldsymbol{w}) = \frac{1}{n} \sum_{i=1}^{n} \mathcal{L}_i(\boldsymbol{w}), \quad \text{where } \mathcal{L}_i(\boldsymbol{w}) = \mathbb{E}_{\boldsymbol{\xi}_i}[\mathcal{L}(\boldsymbol{w}; \boldsymbol{\xi}_i)], i \in [n], \tag{1}$$

where $\boldsymbol{w}$ is the model parameter, $\mathcal{L}_i$ is the loss function on the $i$-th client, $\boldsymbol{\xi}_i$ is the data distribution on the $i$-th client, and $\mathcal{L}(\boldsymbol{w}; \boldsymbol{\xi})$ is the loss function.

In the $t$-th communication round, the server distributes the parameter $\boldsymbol{w}^t$ to the clients. Each client $i$ conducts several epochs of local training on local data to obtain the updated local parameter $\boldsymbol{w}_i^t$. Then, client $i$ computes the local gradient $\boldsymbol{g}_i^t$ as follows and sends it to the server.

$$\boldsymbol{g}_i^t = \boldsymbol{w}^t - \boldsymbol{w}_i^t. \tag{2}$$

Finally, the server collects the local gradients and uses the average gradient to update the global model.

$$\boldsymbol{w}^{t+1} = \boldsymbol{w}^t - \boldsymbol{g}^t, \quad \boldsymbol{g}^t = \frac{1}{n} \sum_{i=1}^{n} \boldsymbol{g}_i^t. \tag{3}$$

This process is repeated until the global model converges or the number of communication rounds reaches the set value $T$.

**Byzantine threat model.** In real-world applications, not all clients in FL systems are honest. In other words, there may exist Byzantine clients in FL systems (Blanchard et al., 2017). Suppose that an adversary controls $f$ Byzantines clients among the total $n$ clients. Let $\mathcal{B} \in [n]$ denote set of Byzantine clients and $\mathcal{H} = [n] \setminus \mathcal{B}$ denote the set of honest clients. In the presence of Byzantine clients, the uploaded message of client $i$ in the $t$-th communication round is

$$\boldsymbol{g}_i^t = \begin{cases} \boldsymbol{w}_i^{t+1} - \boldsymbol{w}^t, & i \in \mathcal{H}, \\ *, & i \in \mathcal{B}, \end{cases} \tag{4}$$

where $*$ represents arbitrary value.

**Robust AGRs.** Most solutions replace the averaging step by a robust alternative to defend against Byzantine attacks. More specifically, the server aggregates the gradients and updates the global model as follows.

$$\boldsymbol{w}^{t+1} = \boldsymbol{w}^t - \hat{\boldsymbol{g}}^t, \quad \hat{\boldsymbol{g}}^t = \mathcal{A}(\boldsymbol{g}_1^t, \ldots, \boldsymbol{g}_n^t), \tag{5}$$

where $\hat{\boldsymbol{g}}^t$ is the aggregated gradient, and $\mathcal{A}$ is a robust AGR, e.g., Multi-Krum (Blanchard et al., 2017), Bulyan (Guerraoui et al., 2018).

For notation simplicity, we omit the superscript $t$ of the gradient symbols when there is no ambiguity in the rest of this paper.

# 4    FAILURES OF EXISTING ROBUST AGRS IN THE NON-IID SETTING

Most robust AGRs focus on Byzantine robustness in the IID setting (Blanchard et al., 2017; Guerraoui et al., 2018). When the data is non-IID, the performance of these robust AGRs drop drastically (Shejwalkar & Houmansadr, 2021; Karimireddy et al., 2022). In order to understand the root cause of this performance drop, we perform an experimental study on various robust AGRs. Particularly, we examine the behaviors of robust AGRs under the attack of $20\%$ Byzantines in both IID and non-IID settings on CIFAR-10 (Krizhevsky et al., 2009) in Figure 1. More detailed setups are covered in Appendix A. A close inspection reveals that AGRs of different types demonstrate different failures in the non-IID setting. As mentioned earlier in Sec. 2, most robust AGRs fall under the umbrella of either *conservative* AGRs or *radical* AGRs. Next, we choose two representative AGRs (Bulyan and Multi-Krum) from both types, and summarize how they fail in the non-IID setting.

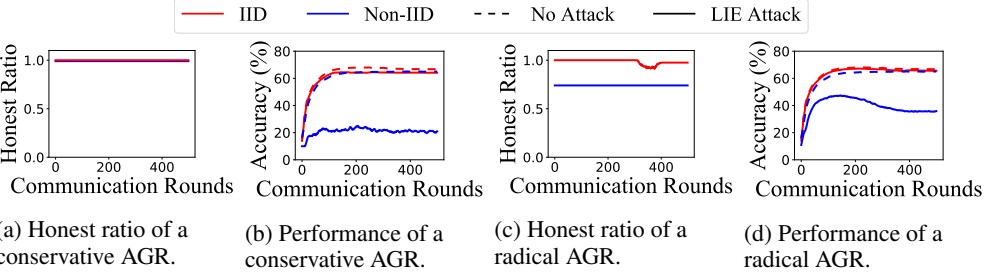

(a) Honest ratio of a conservative AGR.

(b) Performance of a conservative AGR.

(c) Honest ratio of a radical AGR.

(d) Performance of a radical AGR.

Figure 1: The behaviors of a conservative AGR (Bulyan) and a radical AGR (Multi-Krum) under the attack of $20\%$ Byzantines in both IID and non-IID settings on CIFAR-10 dataset. More detailed setups are covered in Appendix A. The dotted lines represent the performance without Byzantines.

**Integrity failure of a conservative AGR.** We take a closer look at a representative conservative AGR – Bulyan (Guerraoui et al., 2018). Specifically, we consider an indicator called *honest ratio* – the ratio of selected honest client number to all selected client number (# selected honest clients / # selected clients) of a robust AGR in each communication round. A higher honest ratio suggests a higher proportion of honest gradients among the gradients aggregated by AGR. In particular, honest ratio 1 (0) suggests that all gradients that the AGR aggregates are honest (Byzantine). In Figure 1(a), we report the honest ratio of the conservative AGR (Bulyan) in both IID and non-IID settings. The results show that in both settings, all the gradients aggregated by the conservative AGR are honest, which demonstrates the strong Byzantine filtering ability of a conservative AGR. Unfortunately, we find that Byzantine filtering is not enough in the non-IID setting. The results in Figure 1(b) illustrate that the accuracy is significantly lower in the non-IID setting. This performance degradation implies a sharp deviation of the aggregated gradient from the *optimal* gradient (the average of honest gradients) in the non-IID setting. Honest gradients are heterogeneous when the data is non-IID (Li et al., 2020; Wang et al., 2021; Karimireddy et al., 2020). As a result, aggregating only partial honest gradients will deviate the aggregated gradient from the optimal gradient, and eventually lead to ineffectual models. Therefore, in addition to Byzantine filtering, *it is also crucial to incorporate sufficient honest gradients into aggregation in the non-IID setting*.

**Identification failure of a radical AGR.** We closely examine a typical radical AGR – Multi-Krum (Blanchard et al., 2017). In Figure 1(c), we demonstrate the honest ratio of the radical AGR in both IID and non-IID settings. As shown in the figure, the radical AGR succeeds in identifying honest gradients for aggregation in the IID setting but fails in the non-IID setting. The critical reason behind is that while the curse of dimensionality (Guerraoui et al., 2018) can be provably addressed in the IID setting, it will be aggravated and intractable in the non-IID setting. When the data is non-IID, Byzantines can easily exploit the curse of dimensionality to compromise radical AGRs, thus degrade the utility of global model as shown in Figure 1(d). Therefore, *it is critical to overcome the aggravated curse of dimensionality in the non-IID setting*.

As analyzed above, most robust AGRs suffer from identification failure and integrity failure due to gradient heterogeneity and the aggravated curse of dimensionality in the non-IID setting. As a result, they fail to stop the aggregated gradient deviating from the optimal gradient, which leads to unsatisfactory performance in the non-IID setting.

## 5 PROPOSED METHOD

Our observations in Sec. 4 clearly motivate the need for a more robust defense to defeat Byzantine attacks in the non-IID setting. Inspired by these observations, we propose a novel GrAdient decomposItioN method called GAIN, which consists of three stages as follows.

**Decomposition.** First, GAIN decomposes the gradients for gradient identification. The decomposition is specified by a partition of set $[d]$, where $d$ is the dimension of gradients. Let $\{\mathcal{J}_1, \ldots \mathcal{J}_p\}$ denote the partition, where $p$ is the number of groups. Particularly, the partition satisfies:

$$\mathcal{J}_q \neq \emptyset, \quad \forall q \in [p] \quad \text{and} \quad [d] = \bigcup_{q=1}^{p} \mathcal{J}_q, \quad \text{and} \quad \mathcal{J}_q \bigcap \mathcal{J}_{q'} = \emptyset, \quad \forall q, q' \in [p], q \neq q', \quad (6)$$

where $\emptyset$ represents the empty set, $\bigcup$ respresents the union of sets, and $\bigcap$ respresents the intersection of sets. Each gradient $\boldsymbol{g}_i$ is correspondingly decomposed into $p$ sub-vectors as follows.

$$\boldsymbol{g}_i^{(q)} = [\boldsymbol{g}_i]_{\mathcal{J}_q}, \quad i \in [n], q \in [p], \quad (7)$$

where $\boldsymbol{g}_i^{(q)}$ is the $q$-th sub-vector of gradient $\boldsymbol{g}_i$.

**Identification.** Then, GAIN applies any robust AGR $\mathcal{A}$ to each group of sub-vectors corresponding to $\mathcal{J}_q$:

$$\hat{\boldsymbol{g}}^{(q)} = \mathcal{A}(\boldsymbol{g}_1^{(q)}, \ldots, \boldsymbol{g}_n^{(q)}), \quad q \in [p], \quad (8)$$

where $\hat{\boldsymbol{g}}^{(q)}$ is the aggregation result of group $q$. By performing aggregation on groups of low-dimensional sub-vectors, GAIN can circumvent the curse of dimensionality, thus avoid the identification failure discussed in Sec. 4. In other words, $\hat{\boldsymbol{g}}^{(q)}$ can get rid of Byzantines.

Note that $\hat{\boldsymbol{g}}^{(q)}$ may still suffer from deviation due to the integrity failure of the AGR $\mathcal{A}$ as illustrated in Sec. 4. Therefore, it is inappropriate to directly use the aggregation results $\{\hat{\boldsymbol{g}}^{(q)}, q \in [p]\}$ as the final output. Instead, we use $\hat{\boldsymbol{g}}^{(q)}$ as an honest reference to compute identification scores for each client as follows.

$$s_i^{(q)} = \|\boldsymbol{g}_i^{(q)} - \hat{\boldsymbol{g}}^{(q)}\|, \quad i \in [n], q \in [p]. \quad (9)$$

Since the group-wise aggregation result $\hat{\boldsymbol{g}}^{(q)}$ can get rid of Byzantines, the identification score $s_i^{(q)}$ can provably characterize the potential for the $\boldsymbol{g}_i^{(q)}$ being a sub-vector of an honest gradient. Then, GAIN collects the identification scores from all groups and computes the final aggregation result. In particular, the final identification score $s_i$ of each client is composed of its identification scores received from all groups as follows.

$$s_i = \sum_{q=1}^{p} s_i^{(q)}, \quad i \in [n]. \quad (10)$$

**Aggregation.** To avoid integrity failure, GAIN selects total $n - f$ gradients with the lowest identification scores for aggregation. Let $\mathcal{I}$ denote the index set of selected gradients, then the average of selected gradients is output as the final aggregation result as follows:

$$\hat{\boldsymbol{g}} = \frac{1}{n-f} \sum_{i \in \mathcal{I}} \boldsymbol{g}_i. \quad (11)$$

Note that in the first stage (Decomposition) of GAIN, $\mathcal{A}$ could be any $c$-resilient AGR (Definition 1). The key difference lies in that all the existing robust AGRs (Multi-Krum, Bulyan, etc) directly operate on the original gradients before aggregation; instead we propose to apply robust AGRs on the decomposed gradient, followed by identification before aggregation. In this way, we can help enhance the identification ability and integrity of the current robust AGRs that satisfy the $c$-resilient property (Definition 1) in the non-IID setting. Detailed theoretical analysis and empirical support can be referred to Sec. 6 and Sec. 7 respectively.

# 6 THEORETICAL ANALYSIS

In this section, we provide a theoretical convergence analysis for our GAIN.

We analyze a popular FL model widely considered by Karimireddy et al. (2021; 2022); Acharya et al. (2022). In particular, each local gradient is computed by SGD as follows.

$$\boldsymbol{g}_i^t = \nabla \mathcal{L}(\boldsymbol{w}^t; \boldsymbol{\xi}_i^t), \quad i \in [n], \tag{12}$$

where $\boldsymbol{\xi}_i^t$ represents a minibatch uniformly sampled from the local data distribution $\boldsymbol{\xi}_i$ in the $t$-th communication round, and $\nabla \mathcal{L}(\boldsymbol{w}^t, \boldsymbol{\xi}_i^t)$ represents the gradient of loss over the minibatch $\boldsymbol{\xi}_i^t$.

We make the following assumptions, which are standard in FL (Karimireddy et al., 2021; 2022; Acharya et al., 2022).

**Assumption 1** (Unbiased Estimator). *The stochastic gradients sampled from any local data distribution are unbiased estimators of local gradients over $\mathbb{R}^d$ for all clients, i.e.,*

$$\mathbb{E}_{\boldsymbol{\xi}_i^t}[\nabla \mathcal{L}(\boldsymbol{w}; \boldsymbol{\xi}_i^t)] = \nabla \mathcal{L}_i(\boldsymbol{w}), \quad \forall \boldsymbol{w} \in \mathbb{R}^d, i \in [n], t \in \mathbb{N}^+. \tag{13}$$

**Assumption 2** (Bounded Variance). *The variance of stochastic gradients sampled from any local data distribution is uniformly bounded over $\mathbb{R}^d$ for all clients, i.e., there exists $\sigma \geq 0$ such that*

$$\mathbb{E}\|\nabla \mathcal{L}(\boldsymbol{w}; \boldsymbol{\xi}_i^t) - \nabla \mathcal{L}_i(\boldsymbol{w})\|^2 \leq \sigma^2, \quad \forall \boldsymbol{w} \in \mathbb{R}^d, i \in [n], t \in \mathbb{N}^+. \tag{14}$$

**Assumption 3** (Gradient Dissimilarity). *The difference between the local gradients and the global gradient is uniformly bounded over $\mathbb{R}^d$ for all clients, i.e., there exists $\kappa \geq 0$ such that*

$$\|\nabla \mathcal{L}_i(\boldsymbol{w}) - \nabla \mathcal{L}(\boldsymbol{w})\|^2 \leq \kappa^2, \quad \forall \boldsymbol{w} \in \mathbb{R}^d, i \in [n]. \tag{15}$$

We consider arbitrary non-convex loss function $\mathcal{L}(\cdot)$ that satisfies the following Lipschitz condition. This condition is widely applied in convergence analysis of Byzantine-robust federated learning (Karimireddy et al., 2022; Allen-Zhu et al., 2020; El-Mhamdi et al., 2021).

**Assumption 4** (Lipschitz Smoothness). *The loss function is L-Lipschitz smooth with respect over $\mathbb{R}^d$, i.e.,*

$$\|\nabla \mathcal{L}(\boldsymbol{w}) - \nabla \mathcal{L}(\boldsymbol{w}')\| \leq \|\boldsymbol{w} - \boldsymbol{w}'\|, \quad \forall \boldsymbol{w}, \boldsymbol{w}' \in \mathbb{R}^d. \tag{16}$$

Assumption 1 establishes the unbiased property of stochastic gradient. Assumption 2 bounds the variance of the stochastic gradients within a client. And Assumption 3 is a common measure of the non-IID level in federated learning (Data & Diggavi, 2021; Karimireddy et al., 2020; 2022).

We further establish the Byzantine resilience of the base AGR $\mathcal{A}$.

**Definition 1** ($c$-resilient AGR). *Let $\mathcal{A}$ be an AGR. If for any input $\{\boldsymbol{x}_1, \ldots, \boldsymbol{x}_n\}$ such that there exists a set $\mathcal{H} \in [n]$ of size at least $|\mathcal{H}| > n/2$ that satisfies:*

$$\mathbb{E}\|\boldsymbol{x}_i - \boldsymbol{x}_{i'}\|^2 \leq \rho^2, \quad \forall i, i' \in \mathcal{H}, \tag{17}$$

*the output of $\mathcal{A}$ satisfies:*

$$\mathbb{E}\|\mathcal{A}(\boldsymbol{x}_1, \ldots, \boldsymbol{x}_n) - \boldsymbol{x}\|^2 \leq c\rho^2, where \ \boldsymbol{x} = \frac{1}{|\mathcal{H}|} \sum_{h \in \mathcal{H}} \boldsymbol{x}_h, \tag{18}$$

*then the AGR $\mathcal{A}$ is called $c$-resilient.*

In fact, most popular AGRs (Blanchard et al., 2017; Guerraoui et al., 2018; Karimireddy et al., 2021; 2022) are shown to satisfy this $c$-resilient definition (Farhadkhani et al., 2022).

We show that given any $c$-resilient base AGR $\mathcal{A}$, our GAIN can help the global model to reach a better parameter point.

**Proposition 1.** *Suppose Assumptions 1 to 4 hold, and let $\eta = 1/2L$. Given a $c$-resilient robust AGR $\mathcal{A}$, we start from $\boldsymbol{w}^0$ and run GAIN for $T$ communication rounds, it satisfies*

$$\mathcal{L}(\boldsymbol{w}^0) \geq \frac{3}{16L} \sum_{t=1}^{T} (\|\nabla \mathcal{L}(\boldsymbol{w}^t)\|^2 - e^2), \tag{19}$$

*where*

$$e^2 = \mathcal{O}(\frac{f^2}{(n-f)^2}(\kappa^2 + \sigma^2)(1 + c^2 + \frac{1}{n-f})(1 + \frac{n}{p})). \tag{20}$$

Please refer to Appendix B for the proof. From one hand, Proposition 1 provides an upper bound for the sum of gradient norms in presence of Byzantine gradients. Equation (19) indicates that as the number of communication rounds increases, we can find an approximate optimal parameter $w$ with $\|\nabla\mathcal{L}(w)\| \leq e$. Furthermore, as the number of sub-vectors $p$ increases, the approximation becomes better, i.e., $e^2$ decreases, which validates the efficacy of our method. From another hand, Proposition 1 characterizes the fundamental difficulties of Byzantine-robust federated learning in the non-IID setting. The negative term $-e^2$ on the RHS implies that FL may never achieve a convergence point. By contrast, the global model may wander among sub-optimal points. What's more, even after reaching the convergence point, the global model may step to a sub-optimal in the next communication round. A detailed comparison of the convergence rate between our method and recent works is presented in Appendix B.2.

# 7 EXPERIMENTS

## 7.1 EXPERIMENTAL SETUPS

**Datasets.** Our experiments are conducted on three real-world datasets: CIFAR-10 (Krizhevsky et al., 2009), CIFAR-100 (Krizhevsky et al., 2009), a subset of ImageNet (Russakovsky et al., 2015) refered as ImageNet-12 and FEMNIST (Caldas et al., 2018). CIFAR-10 dataset consists of 60,000 32×32 color images in 10 classes, with 6,000 images per class. There are 50,000 training images and 10,000 test images in CIFAR-10 dataset. CIFAR-100 dataset consists of 60,000 32×32 color images in 100 classes, with 600 images per class. There are 50,000 training images and 10,000 test images in CIFAR-100 dataset. ImageNet-12 consists of 15,600 color images in 12 classes, with 1,300 images per class. There are 12,480 training images and 3,120 test images in this subset of ImageNet. FEMNIST consists of 817,851 28×28 gray-scale images in 62 classes. There are 772,066 training images and 857,85 test images in FEMNIST.

For FEMNIST, the data is naturally partitioned into 3,597 clients based on the writer of the digit/character. For each client, we randomly sample a 0.9 portion of data as the training data and let the remaining 0.1 portion of data be the test data following Caldas et al. (2018). Intuitively, the data distribution across different clients is non-IID.

**Evaluated attacks.** We consider six representative attacks BitFlip (Allen-Zhu et al., 2020), LabelFlip (Allen-Zhu et al., 2020), LIE (Baruch et al., 2019), Min-Max (Shejwalkar & Houmansadr, 2021), Min-Sum (Shejwalkar & Houmansadr, 2021) and IPM (Xie et al., 2020). The detailed hyperparameter setting of the attacks are shown in Table 5 in Appendix D.

**Baselines.** We consider 6 robust AGRs: (Blanchard et al., 2017), Bulyan (Guerraoui et al., 2018), Median (Yin et al., 2018), RFA (Pillutla et al., 2019), DnC (Shejwalkar & Houmansadr, 2021), RBTM (El-Mhamdi et al., 2021). Among the above six defenses, Bulyan, Median, and RBTM are conservative, and Multi-Krum, RFA, and DnC are radical. We compare each AGR with its variant with GAIN. The detailed hyperparameter settings of the robust AGRs are listed in Table 6 in Appendix D.

**Evaluation.** We use top-1 accuracy, i.e., the proportion of correctly predicted testing samples to total testing samples, to evaluate the performance of global models. We run each experiment for 5 times and report the mean and standard deviation of the highest accuracy during the training process.

**Other settings.** We utilize AlexNet (Krizhevsky et al., 2017), SqueezeNet (Iandola et al., 2016), ResNet-18 (He et al., 2016) and a four-layer CNN (Caldas et al., 2018) for CIFAR-10, CIFAR-100, ImageNet-12 and FEMNIST, respectively. The number of Byzantine clients of all datasets is set to $f = 0.2 \cdot n$. For the partition of set $\{1, ..., d\}$, we randomly partition $\{1, ..., d\}$ into $p$ disjoint subsets with equal size. Please refer to Table 4 in Appendix D for more details.

## 7.2 EXPERIMENT RESULTS

**Main results.** Table 1 illustrates the results of different defenses against popular attacks on CIFAR-10, CIFAR-100, ImageNet-12 and FEMNIST. From these tables, we observe that:

(1) Integrating current defenses into our GAIN generally outperform all their original versions on all datasets, which verifies the efficacy of our proposed GAIN. For example, GAIN improves the accuracy of Median by 15.93% under Min-Sum attack on CIFAR-10.

(2) The improvement of DnC+GAIN to DnC is relatively mild on CIFAR-10. Our interpretation is that when the dataset is relatively small and simple, DnC is capable of obtaining a rational gradient estimation. Nevertheless, on larger and more complex datasets, i.e., FEMNIST and ImageNet-12, DnC fails to achieve satisfactory performance under Byzantine attacks.

(3) We find that although RFA collapses on FEMNIST, integrating into our GAIN can still improve it to satisfactory performance. Our illustration is that although the aggregated gradient of RFA deviates from the optimal gradient, it can still assist in identifying honest gradients when combined with GAIN. As a result, GAIN-RFA is still effective on FEMNIST.

(4) Note that the improvement of GAIN on conservative methods is greater. We contribute this phenomenon to the gradient heterogeneity due to non-IID data. Excluding honest gradients deviates the aggregated gradients from the average of honest gradients, thus degrading the performance of conservative methods. When the non-IID degree increases, the gradient heterogeneity increases. As a result, the impact of excluding honest gradients may even be larger than incorporating Byzantine gradients. Therefore, the improvement on the conservative AGRs is greater.

Table 1: Accuracy (mean±std) of different defenses under 6 attacks on CIFAR-10, ImageNet-12, FEMNIST, and CIFAR-100.

| Dataset | CIFAR-10 | | | | | | CIFAR-100 | | | | | |
|---|---|---|---|---|---|---|---|---|---|---|---|---|
| Attack | BitFlip | LabelFlip | LIE | Min-Max | Min-Sum | IPM | BitFlip | LabelFlip | LIE | Min-Max | Min-Sum | IPM |
| Multi-Krum | 43.19 ± 0.38 | 43.90 ± 0.03 | 37.03 ± 1.62 | 39.06 ± 0.07 | 23.68 ± 0.18 | 36.47 ± 0.22 | 34.27 ± 0.28 | 35.57 ± 0.94 | 17.17 ± 0.08 | 16.77 ± 0.78 | 22.89 ± 0.61 | 15.93 ± 2.00 |
| Multi-Krum+GAIN | 59.23 ± 0.55 | 61.47 ± 0.26 | 55.66 ± 0.93 | 49.19 ± 0.72 | 53.59 ± 0.96 | 56.94 ± 3.60 | 42.41 ± 0.58 | 42.55 ± 0.12 | 27.81 ± 0.32 | 31.18 ± 1.48 | 41.33 ± 0.50 | 42.62 ± 1.53 |
| Bulyan | 54.10 ± 0.19 | 55.12 ± 0.14 | 30.58 ± 0.75 | 29.03 ± 1.10 | 46.19 ± 0.92 | 33.88 ± 0.61 | 35.77 ± 0.18 | 42.60 ± 0.07 | 35.41 ± 0.40 | 35.53 ± 1.38 | 39.13 ± 0.12 | 40.27 ± 1.64 |
| Bulyan+GAIN | 59.14 ± 0.01 | 61.21 ± 0.60 | 48.90 ± 0.83 | 48.35 ± 1.58 | 53.74 ± 0.71 | 56.53 ± 1.51 | 42.28 ± 1.61 | 43.77 ± 0.46 | 38.39 ± 0.19 | 36.33 ± 1.51 | 40.73 ± 0.39 | 42.88 ± 0.14 |
| Median | 45.41 ± 0.44 | 51.88 ± 0.62 | 28.75 ± 0.35 | 32.72 ± 0.81 | 37.39 ± 0.90 | 43.21 ± 0.47 | 36.62 ± 0.12 | 41.64 ± 0.76 | 22.75 ± 0.04 | 23.21 ± 0.16 | 30.68 ± 0.26 | 40.98 ± 0.38 |
| Median+GAIN | 59.28 ± 0.24 | 61.24 ± 1.34 | 46.60 ± 0.13 | 49.37 ± 1.13 | 53.32 ± 1.90 | 56.33 ± 0.82 | 42.41 ± 0.66 | 42.62 ± 0.09 | 35.16 ± 1.08 | 36.46 ± 0.10 | 41.08 ± 0.04 | 43.63 ± 2.85 |
| RFA | 49.61 ± 0.31 | 44.35 ± 0.31 | 15.39 ± 0.37 | 16.62 ± 0.83 | 18.22 ± 0.43 | 45.92 ± 0.13 | 21.32 ± 0.84 | 28.76 ± 1.33 | 25.63 ± 0.20 | 26.46 ± 1.83 | 28.33 ± 0.93 | 21.36 ± 0.54 |
| RFA+GAIN | 53.35 ± 0.30 | 62.25 ± 0.56 | 52.69 ± 0.89 | 52.64 ± 1.48 | 56.16 ± 0.91 | 62.26 ± 1.27 | 42.64 ± 0.44 | 42.42 ± 0.25 | 26.30 ± 1.08 | 30.30 ± 0.12 | 41.09 ± 0.66 | 43.45 ± 0.52 |
| DnC | 58.63 ± 1.29 | 60.82 ± 1.56 | 61.07 ± 0.72 | 60.42 ± 0.59 | 53.71 ± 0.96 | 59.99 ± 0.82 | 41.77 ± 0.62 | 42.93 ± 0.07 | 42.95 ± 1.03 | 40.15 ± 0.70 | 40.02 ± 1.07 | 41.23 ± 2.29 |
| DnC+GAIN | 58.96 ± 0.60 | 61.02 ± 0.27 | 61.87 ± 0.51 | 61.04 ± 1.18 | 54.36 ± 1.12 | 57.92 ± 1.71 | 43.35 ± 0.41 | 43.57 ± 1.11 | 43.64 ± 0.11 | 41.66 ± 0.78 | 41.02 ± 1.39 | 43.25 ± 0.43 |
| RBTM | 54.27 ± 1.63 | 59.60 ± 1.76 | 47.67 ± 2.51 | 49.02 ± 0.31 | 50.74 ± 0.06 | 55.27 ± 1.60 | 36.35 ± 0.17 | 42.67 ± 1.55 | 24.06 ± 0.09 | 26.24 ± 1.04 | 36.51 ± 0.40 | 43.12 ± 1.12 |
| RBTM+GAIN | 59.41 ± 0.20 | 60.75 ± 0.19 | 52.10 ± 1.28 | 49.60 ± 0.17 | 53.63 ± 0.58 | 56.65 ± 1.52 | 43.44 ± 0.81 | 43.19 ± 2.65 | 33.14 ± 0.58 | 34.35 ± 0.76 | 41.51 ± 0.93 | 43.20 ± 0.76 |
| Dataset | FEMNIST | | | | | | ImageNet-12 | | | | | |
| Attack | BitFlip | LabelFlip | LIE | Min-Max | Min-Sum | IPM | BitFlip | LabelFlip | LIE | Min-Max | Min-Sum | IPM |
| Multi-Krum | 67.65 ± 0.23 | 57.43 ± 1.25 | 44.58 ± 0.07 | 28.32 ± 0.31 | 29.98 ± 0.45 | 12.26 ± 1.34 | 44.36 ± 1.52 | 34.04 ± 1.69 | 45.38 ± 1.04 | 48.72 ± 0.16 | 57.69 ± 0.30 | 33.14 ± 0.86 |
| Multi-Krum+GAIN | 84.29 ± 1.76 | 85.45 ± 0.40 | 74.76 ± 1.74 | 57.46 ± 0.33 | 70.65 ± 1.35 | 81.46 ± 0.18 | 66.79 ± 1.08 | 63.04 ± 0.14 | 57.15 ± 0.19 | 59.94 ± 0.32 | 64.07 ± 1.38 | 61.92 ± 0.04 |
| Bulyan | 77.58 ± 1.30 | 79.39 ± 2.14 | 56.43 ± 0.45 | 35.10 ± 0.69 | 44.83 ± 1.40 | 5.91 ± 0.17 | 62.28 ± 0.84 | 59.84 ± 1.09 | 48.04 ± 2.22 | 48.97 ± 1.87 | 59.94 ± 0.51 | 60.67 ± 0.07 |
| Bulyan+GAIN | 84.90 ± 0.69 | 83.68 ± 0.76 | 71.43 ± 1.07 | 66.22 ± 0.47 | 71.76 ± 0.99 | 82.97 ± 1.04 | 66.76 ± 0.72 | 62.28 ± 0.32 | 57.44 ± 0.39 | 58.81 ± 0.05 | 65.00 ± 0.08 | 62.76 ± 0.14 |
| Median | 80.25 ± 0.06 | 76.86 ± 1.96 | 64.88 ± 0.23 | 50.67 ± 0.37 | 61.33 ± 0.13 | 71.98 ± 0.77 | 55.93 ± 0.55 | 58.14 ± 0.18 | 46.67 ± 1.01 | 49.07 ± 1.19 | 58.40 ± 0.03 | 43.62 ± 1.72 |
| Median+GAIN | 84.59 ± 0.14 | 85.67 ± 0.48 | 76.19 ± 0.43 | 65.84 ± 0.41 | 70.84 ± 0.86 | 82.18 ± 0.40 | 66.28 ± 0.41 | 62.34 ± 1.10 | 60.74 ± 1.24 | 59.26 ± 0.31 | 64.78 ± 2.10 | 62.24 ± 0.51 |
| RFA | 5.46 ± 0.06 | 5.46 ± 0.01 | 5.46 ± 0.05 | 5.46 ± 0.03 | 5.46 ± 0.02 | 5.59 ± 0.09 | 61.12 ± 1.26 | 61.31 ± 1.68 | 49.49 ± 1.33 | 53.04 ± 0.13 | 61.92 ± 0.67 | 63.97 ± 0.93 |
| RFA+GAIN | 84.86 ± 0.78 | 84.59 ± 0.20 | 69.82 ± 0.33 | 69.18 ± 0.09 | 77.67 ± 1.31 | 86.08 ± 2.51 | 66.92 ± 1.58 | 63.88 ± 0.94 | 61.41 ± 0.02 | 59.42 ± 0.64 | 67.02 ± 0.54 | 66.67 ± 0.38 |
| DnC | 8.90 ± 0.31 | 77.71 ± 0.03 | 78.52 ± 0.28 | 8.29 ± 0.37 | 74.18 ± 0.03 | 74.70 ± 1.57 | 54.94 ± 0.04 | 5.59 ± 0.06 | 58.01 ± 1.52 | 58.11 ± 0.41 | 60.42 ± 1.60 | 59.99 ± 0.50 |
| DnC+GAIN | 84.71 ± 0.39 | 85.39 ± 0.64 | 82.54 ± 0.26 | 74.37 ± 0.50 | 75.41 ± 0.22 | 82.73 ± 1.22 | 65.19 ± 1.63 | 63.01 ± 0.27 | 64.42 ± 0.19 | 65.03 ± 1.23 | 65.38 ± 1.68 | 65.03 ± 0.04 |
| RBTM | 82.57 ± 0.34 | 81.57 ± 1.12 | 59.93 ± 0.20 | 65.20 ± 0.60 | 71.82 ± 0.73 | 76.88 ± 1.75 | 60.06 ± 1.76 | 60.44 ± 0.37 | 55.77 ± 0.82 | 57.50 ± 0.10 | 63.91 ± 0.78 | 56.19 ± 1.05 |
| RBTM+GAIN | 84.89 ± 1.94 | 85.44 ± 0.20 | 73.38 ± 0.31 | 66.24 ± 0.94 | 75.50 ± 1.13 | 82.58 ± 1.85 | 66.99 ± 0.38 | 61.92 ± 1.22 | 59.87 ± 0.72 | 59.81 ± 1.34 | 64.94 ± 0.72 | 63.40 ± 0.97 |

**Number of sub-vectors.** We study the influence of sub-vector number $p$ on the heterogeneous CIFAR-10 dataset. Figure 2 shows the performance and honest ratio of a conservative AGR (Bulyan) and a radical AGR (Multi-Krum) across $p = \{100, 1000\}$. The results show that for both conservative and radical AGRs, GAIN with a larger sub-vector number $p$ can select a higher proportion of honest clients and achieve a better performance. When the sub-vector number $p$ increases, our GAIN can better handle the identification failure and the integrity failure, which corresponds to our theoretical analysis in Sec. 6.

**Results on different levels of non-IID.** We discuss the impact of non-IID level of data distributions. We modify the concentration parameter $\beta$ to change the non-IID level. A smaller $\beta$ implies a higher non-IID level. Table 2 demonstrates the accuracy of different defenses under LIE attack on CIFAR-10 dataset across $\beta = \{0.3, 0.7\}$. Other setups follow the default setup of the main experiments as illustrated in Sec. 7.1 and Appendix D. As shown in Table 2, all the existing AGRs that combined with GAIN achieve better performances than their original versions, which validates that integrating into our GAIN can effectively defend against Byzantine attacks under different non-IID levels. Moreover, when the level of non-IID is higher, the improvement on robust AGRs is more

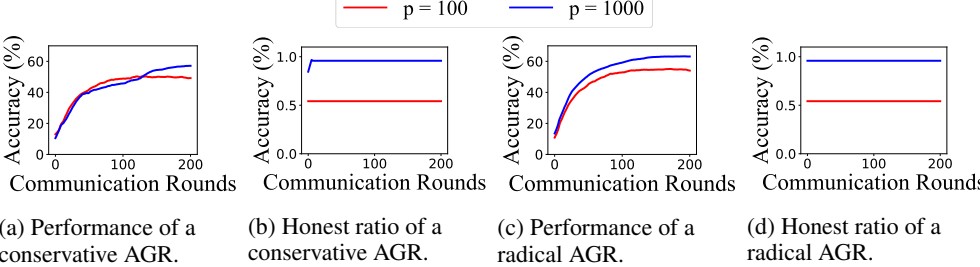

(a) Performance of a conservative AGR.

(b) Honest ratio of a conservative AGR.

(c) Performance of a radical AGR.

(d) Honest ratio of a radical AGR.

Figure 2: The behaviors of a conservative AGR (Bulyan) and a radical AGR (Multi-Krum) across sub-vector number $p = 100, 1000$ under the attack of $20\%$ Byzantines in the non-IID setting on CIFAR-10 dataset. More detailed setups are covered in Appendix D.2.

significant. The results further confirm that our GAIN can overcome the failures aggravated under a higher non-IID level.

Table 2: Accuracy (mean±std) of different defenses against LIE attack under different non-IID levels on CIFAR-10. A smaller $\beta$ implies a higher non-IID level.

| $\beta$ | Multi-Krum | Multi-Krum+GAIN | Bulyan | Bulyan+GAIN | Median | Median+GAIN |
|---|---|---|---|---|---|---|
| 0.3 | $12.19 \pm 1.04$ | $\mathbf{52.80} \pm 0.74$ | $28.16 \pm 0.44$ | $\mathbf{42.81} \pm 0.63$ | $25.62 \pm 0.83$ | $\mathbf{40.97} \pm 0.89$ |
| 0.7 | $31.01 \pm 0.54$ | $\mathbf{55.64} \pm 0.60$ | $44.72 \pm 1.43$ | $\mathbf{51.29} \pm 0.35$ | $34.04 \pm 0.29$ | $\mathbf{53.34} \pm 0.08$ |

| $\beta$ | RFA | RFA+GAIN | DnC | DnC+GAIN | RBTM | RBTM+GAIN |
|---|---|---|---|---|---|---|
| 0.3 | $20.08 \pm 0.13$ | $\mathbf{48.77} \pm 0.84$ | $59.99 \pm 1.81$ | $\mathbf{60.21} \pm 0.62$ | $37.67 \pm 0.18$ | $\mathbf{49.27} \pm 0.05$ |
| 0.7 | $18.11 \pm 0.24$ | $\mathbf{53.25} \pm 1.41$ | $62.15 \pm 0.73$ | $\mathbf{62.48} \pm 0.52$ | $48.43 \pm 0.22$ | $\mathbf{52.25} \pm 1.16$ |

**Results on different number of Byzantine clients.** We also conduct experiments across different number of Byzantine clients. Other setups follow the default setup of the main experiments in Sec. 7.1 and Appendix D. Table 3 demonstrates the results of different defenses under LIE attack across $f = \{5, 15\}$ Byzantine clients on CIFAR-10 dataset. As shown in Table 3, our GAIN outperforms the corresponding baselines across all Byzantine client numbers.

Table 3: Accuracy (mean±std) of different defenses against LIE attack with different Byzantine client numbers $f = \{5, 15\}$ on CIFAR-10. The number of total clients is $n = 50$.

| $f$ | Multi-Krum | Multi-Krum+GAIN | Bulyan | Bulyan+GAIN | Median | Median+GAIN |
|---|---|---|---|---|---|---|
| 5 | $41.65 \pm 1.78$ | $\mathbf{61.24} \pm 0.01$ | $56.28 \pm 1.44$ | $\mathbf{58.27} \pm 0.17$ | $46.91 \pm 1.36$ | $\mathbf{57.69} \pm 1.81$ |
| 15 | $10.00 \pm 0.00$ | $\mathbf{34.70} \pm 0.28$ | $10.00 \pm 0.00$ | $\mathbf{31.67} \pm 0.19$ | $18.85 \pm 1.54$ | $\mathbf{30.95} \pm 0.42$ |

| $f$ | RFA | RFA+GAIN | DnC | DnC+GAIN | RBTM | RBTM+GAIN |
|---|---|---|---|---|---|---|
| 5 | $22.37 \pm 1.00$ | $\mathbf{58.06} \pm 1.29$ | $62.27 \pm 0.04$ | $\mathbf{63.14} \pm 0.20$ | $55.92 \pm 0.10$ | $\mathbf{59.72} \pm 0.16$ |
| 15 | $16.16 \pm 0.14$ | $\mathbf{40.37} \pm 0.26$ | $57.28 \pm 1.37$ | $\mathbf{60.14} \pm 1.64$ | $34.93 \pm 1.36$ | $\mathbf{35.78} \pm 1.51$ |

**Results on different number of clients.** We further analyze the efficacy of our GAIN under different number of clients. Detailed setups can be found in Appendix F. And the experimental results are shown in Table 7 in Appendix F. All results demonstrate that AGRs that combine with our GAIN consistently outperform all their original versions, which validates that integrating with our GAIN can effectively defend against Byzantine across different number of clients.

## 8 CONCLUSION

In this work, we identify two root causes of performance degradation of robust AGRs in the non-IID setting. The first cause is the integrity failure of conservative AGRs. Conservative AGRs aggregate only few honest gradients, which is unreliable due to the gradient heterogeneity in the non-IID setting. The second cause is the identification failure of radical AGRs. Radical AGRs inevitably introduce Byzantine gradients into aggregation due to the curse of dimensionality aggravated by the non-IIDness. Both failures result in a sharp deviation of the aggregated gradient. Motivated by the above discoveries, we propose a novel GrAdient decomposItioN (GAIN) method that can be combined with most existing defenses and overcome both failures. We also provide convergence analysis for integrating existing robust AGRs into GAIN. Empirical studies on three real-world datasets justify the efficacy of our proposed GAIN.

ETHICS STATEMENT

In this paper, our studies are not related to human subjects, practices to dataset releases, discrimination/bias/fairness concerns, and also do not have legal compliance or research integrity issues. Our work is proposed to achieve Byzantine robustness when applying federated learning to real-world applications. In this case, if federated learning is applied for good, we believe our proposed method will not cause any ethical problems or pose any negative societal impacts.

REPRODUCIBILITY STATEMENT

The implementation code is provided in Supplementary Materials. All datasets and the code platform (PyTorch) we use are public. Detail experiment setups are provided in the Appendices A and D.

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

## A   SETUPS FOR EXPERIMENTS IN SEC. 4

The experiments are conducted on CIFAR-10 (Krizhevsky et al., 2009).

For both IID and non-IID settings, the number of client is set to $n = 50$. For IID data distribution, all 50,000 samples are randomly partitioned into 50 clients each containing 1,000 samples. For non-IID data distribution, the samples are partitioned in a Dirichlet manner with concentration parameter $\beta = 0.5$. Please refer to Sec. 7.1 for the details of Dirichlet partition.

The number of Byzantine clients is set to $f = 10$. LIE (Baruch et al., 2019) attack with $z = 1.5$ considered.

We use AlexNet (Krizhevsky et al., 2017) as the model architecture. The number of communication rounds is set to 500. In each communication round, all client participate in the training.

For local training, the number of local epochs is set to 1, batch size is set to 64, the optimizer is set to SGD. For SGD optimizer, learning rate is set to 0.1, momentum is set to 0.5, weight decay coeffecient is set to 0.0001. We also adopt gradient clipping with clipping norm 2.

Two defenses are considered: a radical AGR Multi-Krum (Blanchard et al., 2017) and a conservative AGR Bulyan (Guerraoui et al., 2018).

## B   PROOF FOR PROPOSITION 1

Here we restate the assumptions and the proposition for the integrity of this section.

**Assumption 1** (Unbiased Estimator). *The stochastic gradients sampled from any local data distribution are unbiased estimators of local gradients over $\mathbb{R}^d$ for all clients, i.e.,*

$$\mathbb{E}_{\boldsymbol{\xi}_i^t}[\nabla \mathcal{L}(\boldsymbol{w}; \boldsymbol{\xi}_i^t)] = \nabla \mathcal{L}_i(\boldsymbol{w}), \quad \forall \boldsymbol{w} \in \mathbb{R}^d, i \in [n], t \in \mathbb{N}^+. \tag{13}$$

**Assumption 2** (Bounded Variance). *The variance of stochastic gradients sampled from any local data distribution is uniformly bounded over $\mathbb{R}^d$ for all clients, i.e., there exists $\sigma \geq 0$ such that*

$$\mathbb{E}\|\nabla \mathcal{L}(\boldsymbol{w}; \boldsymbol{\xi}_i^t) - \nabla \mathcal{L}_i(\boldsymbol{w})\|^2 \leq \sigma^2, \quad \forall \boldsymbol{w} \in \mathbb{R}^d, i \in [n], t \in \mathbb{N}^+. \tag{14}$$

**Assumption 3** (Gradient Dissimilarity). *The difference between the local gradients and the global gradient is uniformly bounded over $\mathbb{R}^d$ for all clients, i.e., there exists $\kappa \geq 0$ such that*

$$\|\nabla \mathcal{L}_i(\boldsymbol{w}) - \nabla \mathcal{L}(\boldsymbol{w})\|^2 \leq \kappa^2, \quad \forall \boldsymbol{w} \in \mathbb{R}^d, i \in [n]. \tag{15}$$

**Assumption 4** (Lipschitz Smoothness). *The loss function is $L$-Lipschitz smooth with respect over $\mathbb{R}^d$, i.e.,*

$$\|\nabla \mathcal{L}(\boldsymbol{w}) - \nabla \mathcal{L}(\boldsymbol{w}')\| \leq \|\boldsymbol{w} - \boldsymbol{w}'\|, \quad \forall \boldsymbol{w}, \boldsymbol{w}' \in \mathbb{R}^d. \tag{16}$$

**Definition 1** ($c$-resilient AGR). *Let $\mathcal{A}$ be an AGR. If for any input $\{\boldsymbol{x}_1, \ldots, \boldsymbol{x}_n\}$ such that there exists a set $\mathcal{H} \in [n]$ of size at least $|\mathcal{H}| > n/2$ that satisfies:*

$$\mathbb{E}\|\boldsymbol{x}_i - \boldsymbol{x}_{i'}\|^2 \leq \rho^2, \quad \forall i, i' \in \mathcal{H}, \tag{17}$$

*the output of $\mathcal{A}$ satisfies:*

$$\mathbb{E}\|\mathcal{A}(\boldsymbol{x}_1, \ldots, \boldsymbol{x}_n) - \boldsymbol{x}\|^2 \leq c\rho^2, \text{where } \boldsymbol{x} = \frac{1}{|\mathcal{H}|} \sum_{h \in \mathcal{H}} \boldsymbol{x}_h, \tag{18}$$

*then the AGR $\mathcal{A}$ is called $c$-resilient.*

**Proposition 1.** *Suppose Assumptions 1 to 4 hold, and let $\eta = 1/2L$. Given a $c$-resilient robust AGR $\mathcal{A}$, we start from $\boldsymbol{w}^0$ and run GAIN for $T$ communication rounds, it satisfies*

$$\mathcal{L}(\boldsymbol{w}^0) \geq \frac{3}{16L} \sum_{t=1}^{T} (\|\nabla \mathcal{L}(\boldsymbol{w}^t)\|^2 - e^2), \tag{19}$$

*where*

$$e^2 = \mathcal{O}(\frac{f^2}{(n-f)^2}(\kappa^2 + \sigma^2)(1 + c^2 + \frac{1}{n-f})(1 + \frac{n}{p})). \tag{20}$$

## B.1 KEY LEMMA AND PROOF

Before starting the proof of the main proposition, we first state and prove the following lemma.

**Lemma 1** (Estimation error). *Suppose Assumptions 1 to 3 hold. Given a $c$-resilient robust AGR $\mathcal{A}$, for any $\varepsilon > 0$, with probability at least $1 - \varepsilon$, where $p$ is the number of sub-vectors in GAIN, the aggregated gradient $\hat{g}$ of GAIN is an unbiased estimator of the optimal gradient $\bar{g} = \nabla \mathcal{L}(\boldsymbol{w})$ with bounded variance.*

$$\mathbb{E}\hat{\boldsymbol{g}} = \bar{\boldsymbol{g}}, \quad \mathrm{Var}[\hat{\boldsymbol{g}}] \leq \frac{\sigma^2}{n - f}, \tag{21}$$

*when $\mathbb{E}\|\boldsymbol{g}_b - \bar{\boldsymbol{g}}\| = \Omega(\kappa \cdot (1 + c + \sqrt{n/p\varepsilon}(1 + \sqrt{c})) + \sigma \cdot (1 + c + 1/\sqrt{n - f} + \sqrt{n/p\varepsilon}(1 + \sqrt{c} + 1/\sqrt{n - f}))$.*

We state and prove the following lemma for the proof of Lemma 1.

**Lemma 2.** *For any random vector $\boldsymbol{X}$, we have*

$$\mathrm{Var}[\|\boldsymbol{X}\|] \leq \mathbb{E}\|\boldsymbol{X} - \mathbb{E}\boldsymbol{X}\|^2. \tag{22}$$

*Proof.* From the definition of variance, we have

$$\mathrm{Var}[\|\boldsymbol{X}\|] = \mathbb{E}(\|\boldsymbol{X}\| - \mathbb{E}\|\boldsymbol{X}\|)^2 \tag{23}$$

$$= \mathbb{E}(\|\boldsymbol{X}\| - \|\mathbb{E}\boldsymbol{X}\|)^2 - (\|\mathbb{E}\boldsymbol{X}\| - \mathbb{E}\|\boldsymbol{X}\|)^2 \tag{24}$$

$$\leq \mathbb{E}(\|\boldsymbol{X}\| - \|\mathbb{E}\boldsymbol{X}\|)^2 \tag{25}$$

$$\leq \mathbb{E}\|\boldsymbol{X} - \mathbb{E}\boldsymbol{X}\|^2. \tag{26}$$

The second inequality comes from triangular inequality. $\square$

Equipped with Lemma 2, we start the formal proof for Lemma 1.

*Proof.* For all honest clients $i, j \in \mathcal{H}$, parameter group $q \in [p]$, we have

$$\mathbb{E}\|\boldsymbol{g}_i^{(q)} - \boldsymbol{g}_j^{(q)}\|^2 \tag{27}$$

$$= \mathbb{E}\|(\boldsymbol{g}_i^{(q)} - \bar{\boldsymbol{g}}_i^{(q)}) + (\bar{\boldsymbol{g}}_i^{(q)} - \bar{\boldsymbol{g}}^{(q)}) + (\bar{\boldsymbol{g}}^{(q)} - \bar{\boldsymbol{g}}_j^{(q)}) + (\bar{\boldsymbol{g}}_j^{(q)} - \boldsymbol{g}_j^{(q)})\|^2 \tag{28}$$

$$\leq 4\mathbb{E}[\|\boldsymbol{g}_i^{(q)} - \bar{\boldsymbol{g}}_i^{(q)}\|^2 + \|\bar{\boldsymbol{g}}_i^{(q)} - \bar{\boldsymbol{g}}^{(q)}\|^2 + \|\bar{\boldsymbol{g}}^{(q)} - \bar{\boldsymbol{g}}_j^{(q)}\|^2 + \|\bar{\boldsymbol{g}}_j^{(q)} - \boldsymbol{g}_j^{(q)}\|^2] \tag{29}$$

$$\leq 8\sigma^2 + 8\kappa^2. \tag{30}$$

Here the first inequality comes from the Cauchy inequality, and the second inequality follows Assumptions 2 and 3. Then according to the Definition 1, we have

$$\mathbb{E}\|\hat{\boldsymbol{g}}^{(q)} - \boldsymbol{g}^{(q)}\|^2 \leq 8c(\sigma^2 + \kappa^2) \tag{31}$$

Then for honest client $h$, the expectation of abnormal score $s_h^{(q)}$ from group $q$ can be bounded as follows.

$$\mathbb{E}[s_h^{(q)}] = \mathbb{E}\|\boldsymbol{g}_h^{(q)} - \hat{\boldsymbol{g}}^{(q)}\| \tag{32}$$

$$\leq \mathbb{E}[\|\boldsymbol{g}_h^{(q)} - \bar{\boldsymbol{g}}_h^{(q)}\| + \|\bar{\boldsymbol{g}}_h^{(q)} - \boldsymbol{g}^{(q)}\| + \|\boldsymbol{g}^{(q)} - \hat{\boldsymbol{g}}^{(q)}\|] \tag{33}$$

$$= \mathbb{E}\|\boldsymbol{g}_h^{(q)} - \bar{\boldsymbol{g}}_h^{(q)}\| + \mathbb{E}\|\bar{\boldsymbol{g}}_h^{(q)} - \boldsymbol{g}^{(q)}\| + \mathbb{E}\|\boldsymbol{g}^{(q)} - \hat{\boldsymbol{g}}^{(q)}\| \tag{34}$$

$$\leq \sqrt{\mathbb{E}\|\boldsymbol{g}_h^{(q)} - \bar{\boldsymbol{g}}_h^{(q)}\|^2} + \sqrt{\mathbb{E}\|\bar{\boldsymbol{g}}_h^{(q)} - \boldsymbol{g}^{(q)}\|^2} + \sqrt{\mathbb{E}\|\boldsymbol{g}^{(q)} - \hat{\boldsymbol{g}}^{(q)}\|^2} \tag{35}$$

$$\leq \sigma + \kappa + 2\sqrt{2}c\sqrt{\sigma^2 + \kappa^2}. \tag{36}$$

Here the first inequality is a result of triangular inequality, the second inequality comes from Cauchy inequality, and the third inequality is a combined result of Equation (31) and Assumptions 2 and 3.

The variance of $s_h^{(q)}$ can also be bounded as follows.

$$\text{Var}[s_h^{(q)}] = \mathbb{E}[(s_h^{(q)})^2] - (\mathbb{E}[s_h^{(q)}])^2 \tag{37}$$

$$\leq \mathbb{E}[(s_h^{(q)})^2] \tag{38}$$

$$= \mathbb{E}\|\boldsymbol{g}_h^{(q)} - \hat{\boldsymbol{g}}^{(q)}\|^2 \tag{39}$$

$$\leq 4\mathbb{E}[\|\boldsymbol{g}_h^{(q)} - \bar{\boldsymbol{g}}_h^{(q)}\|^2 + \|\bar{\boldsymbol{g}}_h^{(q)} - \bar{\boldsymbol{g}}^{(q)}\|^2 + \|\bar{\boldsymbol{g}}^{(q)} - \boldsymbol{g}^{(q)}\|^2 + \|\boldsymbol{g}^{(q)} - \hat{\boldsymbol{g}}^{(q)}\|^2]. \tag{40}$$

Here the second inequality is a result of Cauchy inequality.

We bound $\mathbb{E}\|\bar{\boldsymbol{g}}^{(q)} - \boldsymbol{g}^{(q)}\|^2$ as follows.

$$\mathbb{E}\|\bar{\boldsymbol{g}}^{(q)} - \boldsymbol{g}^{(q)}\|^2 = \mathbb{E}\|\frac{1}{n-f}\sum_{i\in\mathcal{H}}(\bar{\boldsymbol{g}}_i^{(q)} - \boldsymbol{g}_i^{(q)})\|^2 \tag{41}$$

$$= \frac{1}{(n-f)^2}\sum_{i\in\mathcal{H}}\mathbb{E}\|\bar{\boldsymbol{g}}_i^{(q)} - \boldsymbol{g}_i^{(q)}\|^2 \tag{42}$$

$$\leq \frac{1}{(n-f)^2}\sum_{i\in\mathcal{H}}\sigma^2 \tag{43}$$

$$= \frac{\sigma^2}{n-f} \tag{44}$$

Here the second equality comes from the independence of minibatches sampling across different clients, and the first inequality is a result of Assumption 2.

Applying Assumptions 2 and 3 and Equations (31) and (44) to Equation (40), we have

$$\text{Var}[s_h^{(q)}] \leq 4(\sigma^2 + \kappa^2 + \frac{\sigma^2}{n-f} + 8c(\sigma^2 + \kappa^2)) \tag{45}$$

$$= (4 + 32c + \frac{4}{n-f})\sigma^2 + (4 + 32c)\kappa^2. \tag{46}$$

According to Equations (36) and (46), we can bound the expectation and variance of total abnormal score $s_h$ of an honest client $h$.

$$\mathbb{E}[s_h] = \mathbb{E}[\sum_{q=1}^{p} s_h^{(q)}] \leq p(\sigma + \kappa + 2\sqrt{2}c\sqrt{\sigma^2 + \kappa^2}) := A, \tag{47}$$

$$\text{Var}[s_h] = \sum_{q=1}^{p}\text{Var}[s_h^{(q)}] \leq p((4 + 32c + \frac{4}{n-f})\sigma^2 + (4 + 32c)\kappa^2) := B. \tag{48}$$

Here the addictive property of variance is a result of the independence of group abnormal scores $\{s_h^{(q)} \mid q \in [p]\}$, which comes from the independence of components in a gradient (Yang & Schoenholz, 2017).

From Chebyshev's inequality, for any $\Delta_h > 0$ and honest client $h \in [n] \setminus \mathcal{B}$, we have

$$P(s_h < \mathbb{E}[s_h] + \Delta_h) \geq 1 - \frac{\text{Var}[s_h]}{\Delta_h^2}. \tag{49}$$

Consider the expectation of abnormal score $s_b^{(q)}$ from group $q$ for Byzantine client $b \in \mathcal{B}$

$$\mathbb{E}[s_b^{(q)}] = \mathbb{E}\|\boldsymbol{g}_b^{(q)} - \hat{\boldsymbol{g}}^{(q)}\| \tag{50}$$

$$= \mathbb{E}\|(\boldsymbol{g}_b^{(q)} - \bar{\boldsymbol{g}}^{(q)}) - (\hat{\boldsymbol{g}}^{(q)} - \bar{\boldsymbol{g}}^{(q)})\| \tag{51}$$

$$\geq \mathbb{E}[\|\boldsymbol{g}_b^{(q)} - \bar{\boldsymbol{g}}^{(q)}\| - \|\hat{\boldsymbol{g}}^{(q)} - \bar{\boldsymbol{g}}^{(q)}\|] \tag{52}$$

$$\geq \mathbb{E}[\|\boldsymbol{g}_b^{(q)} - \bar{\boldsymbol{g}}^{(q)}\| - (\|\hat{\boldsymbol{g}}^{(q)} - \boldsymbol{g}^{(q)}\| + \|\boldsymbol{g}^{(q)} - \bar{\boldsymbol{g}}^{(q)}\|)] \tag{53}$$

$$\geq \mathbb{E}\|\boldsymbol{g}_b^{(q)} - \bar{\boldsymbol{g}}^{(q)}\| - (\sqrt{\mathbb{E}\|\hat{\boldsymbol{g}}^{(q)} - \bar{\boldsymbol{g}}^{(q)}\|^2} + \sqrt{\mathbb{E}\|\boldsymbol{g}^{(q)} - \bar{\boldsymbol{g}}^{(q)}\|^2}) \tag{54}$$

$$\geq \delta_b - 2\sqrt{2}c\sqrt{\sigma^2 + \kappa^2} - \frac{\sigma}{\sqrt{n-f}} \tag{55}$$

where $\delta_b = \mathbb{E}\|\boldsymbol{g}_b^{(q)} - \bar{\boldsymbol{g}}^{(q)}\|$ is the expected deviation of Byzantine client $b$ from the average of honest gradients. Here the first and second inequalities come from triangular inequality, the third inequality is based on Cauchy inequality, and the 4-th inequality is a combined result of Equations (31) and (44).

The variance of abnormal score $s_b^{(q)}$ can be bounded as follows.

$$\text{Var}[s_b^{(q)}] = \text{Var}[\|\boldsymbol{g}_b^{(q)} - \hat{\boldsymbol{g}}^{(q)}\|] \tag{56}$$

$$\leq \mathbb{E}\|\boldsymbol{g}_b^{(q)} - \hat{\boldsymbol{g}}^{(q)} - \mathbb{E}[\boldsymbol{g}_b^{(q)} - \hat{\boldsymbol{g}}^{(q)}]\|^2 \tag{57}$$

$$= \mathbb{E}\|(\boldsymbol{g}_b^{(q)} - \mathbb{E}\boldsymbol{g}_b^{(q)}) - (\hat{\boldsymbol{g}}^{(q)} - \mathbb{E}\hat{\boldsymbol{g}}^{(q)})\|^2 \tag{58}$$

$$\leq 2\mathbb{E}[\|\boldsymbol{g}_b^{(q)} - \mathbb{E}\boldsymbol{g}_b^{(q)}\|^2 + \|\hat{\boldsymbol{g}}^{(q)} - \mathbb{E}\hat{\boldsymbol{g}}^{(q)}\|^2] \tag{59}$$

$$= 2\mathbb{E}\|\boldsymbol{g}_b^{(q)} - \mathbb{E}\boldsymbol{g}_b^{(q)}\|^2 + 2\mathbb{E}\|\hat{\boldsymbol{g}}^{(q)} - \mathbb{E}\hat{\boldsymbol{g}}^{(q)}\|^2 \tag{60}$$

The first inequality results from Lemma 2, and the second inequality comes from Cauchy inequality. We bound $\|\hat{\boldsymbol{g}}^{(q)} - \mathbb{E}\hat{\boldsymbol{g}}^{(q)})\|$ as follows.

$$\mathbb{E}\|\hat{\boldsymbol{g}}^{(q)} - \mathbb{E}\hat{\boldsymbol{g}}^{(q)}\| = \mathbb{E}\|(\hat{\boldsymbol{g}}^{(q)} - \boldsymbol{g}^{(q)}) + (\boldsymbol{g}^{(q)} - \mathbb{E}\boldsymbol{g}^{(q)}) - \mathbb{E}[\hat{\boldsymbol{g}}^{(q)} - \boldsymbol{g}^{(q)}]\|^2] \tag{61}$$

$$\leq 3\mathbb{E}[\|\hat{\boldsymbol{g}}^{(q)} - \boldsymbol{g}^{(q)}\|^2 + \|\boldsymbol{g}^{(q)} - \mathbb{E}\boldsymbol{g}^{(q)}\|^2 + \|\mathbb{E}[\hat{\boldsymbol{g}}^{(q)} - \boldsymbol{g}^{(q)}]\|^2] \tag{62}$$

$$\leq 6\mathbb{E}\|\hat{\boldsymbol{g}}^{(q)} - \boldsymbol{g}^{(q)}\|^2 + 3\mathbb{E}\|\boldsymbol{g}^{(q)} - \mathbb{E}\boldsymbol{g}^{(q)}\|^2 \tag{63}$$

$$\leq 48(\sigma^2 + \kappa^2) + \frac{3\sigma^2}{n - f} \tag{64}$$

$$= (48 + \frac{3\sigma^2}{n - f})\sigma^2 + 48\kappa^2 \tag{65}$$

Apply Equation (65) to Equation (60), we have

$$\text{Var}[s_b^{(q)}] \leq 2\sigma_b^2 + (96 + \frac{6}{n - f})\sigma^2 + 96\kappa^2, \tag{66}$$

where $\sigma_b^2 = \mathbb{E}\|\boldsymbol{g}_b^{(q)} - \mathbb{E}\boldsymbol{g}_b^{(q)}\|^2$ is the variance.

Similar to Equations (47) and (48), we utilize Equations (55) and (66) to bound the expectation and variance of total abnormal score $s_b$ of a byzantine client $b$.

$$\mathbb{E}[s_b] = \mathbb{E}[\sum_{q=1}^{p} s_b^{(q)}] \geq p(\delta_b - 2\sqrt{2}c\sqrt{\sigma^2 + \kappa^2} - \frac{\sigma}{\sqrt{n - f}}) := C, \tag{67}$$

$$\text{Var}[s_b] = \sum_{q=1}^{p} \text{Var}[s_b^{(q)}] \leq p(2\text{const} + (96 + \frac{6}{n - f})\sigma^2 + 96\kappa^2) := D \tag{68}$$

where $\delta_b = \mathbb{E}\|\boldsymbol{g}_b - \bar{\boldsymbol{g}}\|$ According to Shejwalkar & Houmansadr (2021), $\sigma_b^2$ is bounded, i.e., $\sigma_b^2 \leq$ const.

Similarly, we apply Chebyshev's inequality to the abnormal score of a Byzantine client $b \in \mathcal{B}$.

$$\Pr(s_b \geq \mathbb{E}[s_b] - \Delta_b) \geq 1 - \frac{\text{Var}[s_b]}{\Delta_b^2}, \quad b \in \mathcal{B}. \tag{69}$$

Combine Equations (47) to (49), and take $\Delta_h = (C - A)/(1 + \sqrt{D/B})$, we have

$$\Pr(s_h < \frac{\sqrt{D}A + \sqrt{B}C}{\sqrt{B} + \sqrt{D}}) = \Pr(s_h < A + \Delta_h) \tag{70}$$

$$\geq \Pr(s_h < \mathbb{E}[s_h] + \Delta_h) \tag{71}$$

$$\geq 1 - \frac{\mathrm{Var}[s_h]}{\Delta_h^2} \tag{72}$$

$$\geq 1 - \frac{B}{\Delta_h^2} \tag{73}$$

$$= 1 - \frac{(\sqrt{B} + \sqrt{D})^2}{(C - A)^2}, \tag{74}$$

Combine Equations (67) to (69), and take $\Delta_b = (C - A)/(1 + \sqrt{B/D})$, we have

$$\Pr(s_b \geq \frac{\sqrt{D}A + \sqrt{B}C}{\sqrt{B} + \sqrt{D}}) \geq \Pr(s_b > C - \Delta_b) \tag{75}$$

$$\geq \Pr(s_b > \mathbb{E}[s_b] - \Delta) \tag{76}$$

$$\geq 1 - \frac{\mathrm{Var}[s_b]}{\Delta^2} \tag{77}$$

$$\geq 1 - \frac{D}{\Delta_b^2}, \tag{78}$$

$$= 1 - \frac{(\sqrt{B} + \sqrt{D})^2}{(C - A)^2}, \tag{79}$$

Then consider the probability all the Byzantines are filtered

$$\Pr(\{i_1, \ldots, i_{n-f}\} = \mathcal{H}) \geq \Pr(s_h < \frac{\sqrt{D}A + \sqrt{B}C}{\sqrt{B} + \sqrt{D}}, \forall h \in \mathcal{H}, s_b > \frac{\sqrt{D}A + \sqrt{B}C}{\sqrt{B} + \sqrt{D}}, \forall b \in \mathcal{B}) \tag{80}$$

$$= \prod_{h \in \mathcal{H}} \Pr(s_h < \frac{\sqrt{D}A + \sqrt{B}C}{\sqrt{B} + \sqrt{D}}) \prod_{b \in \mathcal{B}} \Pr(s_b \geq \frac{\sqrt{D}A + \sqrt{B}C}{\sqrt{B} + \sqrt{D}}) \tag{81}$$

$$\geq \prod_{h \in \mathcal{H}} (1 - \frac{(\sqrt{B} + \sqrt{D})^2}{(C - A)^2}) \prod_{b \in \mathcal{B}} (1 - \frac{(\sqrt{B} + \sqrt{D})^2}{(C - A)^2}) \tag{82}$$

$$\geq (1 - \frac{(\sqrt{B} + \sqrt{D})^2}{(C - A)^2})^n \tag{83}$$

$$\geq 1 - n \cdot \frac{(\sqrt{B} + \sqrt{D})^2}{(C - A)^2} \tag{84}$$

Solve $1 - n \cdot (\sqrt{B} + \sqrt{D})^2/(C - A)^2 \geq 1 - \varepsilon/$, we have

$$\mathbb{E}\|\boldsymbol{g}_b - \bar{\boldsymbol{g}}\| \geq (1 + \frac{1}{\sqrt{n-f}})\sigma + \kappa + 4\sqrt{2}c\sqrt{\sigma^2 + \kappa^2} \tag{85}$$

$$+ \sqrt{\frac{n}{p\varepsilon}}(\sqrt{((4 + 32c + \frac{4}{n-f})\sigma^2 + (4 + 32c)\kappa^2)} \tag{86}$$

$$+ \sqrt{(2\mathrm{const} + (96 + \frac{6}{n-f})\sigma^2 + 96\kappa^2))} \tag{87}$$

that is,

$$\mathbb{E}\|\boldsymbol{g}_b - \bar{\boldsymbol{g}}\| = \Omega(\kappa \cdot (1 + c + \sqrt{\frac{n}{p\varepsilon}}(1 + \sqrt{c})) + \sigma \cdot (1 + c + \frac{1}{\sqrt{n-f}} + \sqrt{\frac{n}{p\varepsilon}}(1 + \sqrt{c} + \frac{1}{\sqrt{n-f}}))). \tag{88}$$

$$\square$$

## B.2 PROOF FOR THE MAIN PROPOSITION

*Proof.* According to the Lipschitz property of loss function $\mathcal{L}$, we have

$$\mathcal{L}(\boldsymbol{w}^t) - \mathcal{L}(\boldsymbol{w}^{t+1}) \geq \langle \nabla \mathcal{L}(\boldsymbol{w}^t), \boldsymbol{w}^t - \boldsymbol{w}^{t+1} \rangle - \frac{L}{2}\|\boldsymbol{w}^t - \boldsymbol{w}^{t+1}\|^2. \tag{89}$$

Since $\boldsymbol{w}^t - \boldsymbol{w}^{t+1} = \nabla \mathcal{L}(\boldsymbol{w}^t) + (\hat{\boldsymbol{g}}^t - \nabla \mathcal{L}(\boldsymbol{w}^t))$, we can write Equation (89) as follows

$$
\begin{aligned}
\mathcal{L}(\boldsymbol{w}^t) - \mathcal{L}(\boldsymbol{w}^{t+1}) \geq\ & (\eta - \frac{L}{2}\eta^2)\|\nabla \mathcal{L}(\boldsymbol{w}^t)\|^2 \\
& + (\eta - \frac{L}{2}\eta^2) \langle \nabla \mathcal{L}(\boldsymbol{w}^t), \hat{\boldsymbol{g}}^t - \nabla \mathcal{L}(\boldsymbol{w}^t) \rangle \\
& - \frac{L}{2}\eta^2\|\hat{\boldsymbol{g}}^t - \nabla \mathcal{L}(\boldsymbol{w}^t)\|^2.
\end{aligned}
\tag{90}
$$

Take the expectation on both sides of Equation (90), we have

$$
\begin{aligned}
\mathbb{E}\mathcal{L}(\boldsymbol{w}^t) - \mathcal{L}(\boldsymbol{w}^{t+1}) \geq\ & (\eta - \frac{L}{2}\eta^2)\mathbb{E}\|\nabla \mathcal{L}(\boldsymbol{w}^t)\|^2 \\
& + (\eta - \frac{L}{2}\eta^2)\mathbb{E} \langle \nabla \mathcal{L}(\boldsymbol{w}^t), \hat{\boldsymbol{g}}^t - \nabla \mathcal{L}(\boldsymbol{w}^t) \rangle \\
& - \frac{L}{2}\eta^2\mathbb{E}\|\hat{\boldsymbol{g}}^t - \nabla \mathcal{L}(\boldsymbol{w}^t)\|^2.
\end{aligned}
\tag{91}
$$

We further bound terms $\mathbb{E} \langle \nabla \mathcal{L}(\boldsymbol{w}^t), \hat{\boldsymbol{g}}^t - \nabla \mathcal{L}(\boldsymbol{w}^t) \rangle$ and $\mathbb{E}\|\hat{\boldsymbol{g}}^t - \nabla \mathcal{L}(\boldsymbol{w}^t)\|^2$.

First, we bound term $\mathbb{E} \|\hat{\boldsymbol{g}}^t - \nabla \mathcal{L}(\boldsymbol{w}^t)\|^2$.

For notation simplicity, we define $\tilde{\mathcal{H}}^t = \mathcal{H} \cap \mathcal{I}^t$ and $\tilde{\mathcal{B}}^t = \mathcal{B} \cap \mathcal{I}^t$. Then $\hat{\boldsymbol{g}}^t$ can be written as follows:

$$\hat{\boldsymbol{g}}^t = \frac{1}{n-f}\sum_{i \in \mathcal{I}} \boldsymbol{g}_i^t = \frac{1}{n-f}(\sum_{h \in \tilde{\mathcal{H}}} \boldsymbol{g}_h^t + \sum_{b \in \tilde{\mathcal{B}}} \boldsymbol{g}_b^t) = \frac{\tilde{h}}{n-f}\bar{\boldsymbol{g}}_{\tilde{\mathcal{H}}}^t + \frac{\tilde{f}}{n-f}\bar{\boldsymbol{g}}_{\tilde{\mathcal{B}}}^t. \tag{92}$$

Here $\tilde{h} = |\tilde{\mathcal{H}}^t|$, $\tilde{b} = |\tilde{\mathcal{B}}|$, $\bar{\boldsymbol{g}}_{\tilde{\mathcal{H}}}^t = \sum_{h \in \tilde{\mathcal{H}}} \boldsymbol{g}_h^t / \tilde{h}$, and $\bar{\boldsymbol{g}}_{\tilde{\mathcal{B}}}^t = \sum_{b \in \tilde{\mathcal{B}}} \boldsymbol{g}_b^t / \tilde{b}$.

Term $\mathbb{E} \|\hat{\boldsymbol{g}}^t - \nabla \mathcal{L}(\boldsymbol{w}^t)\|^2$ is then bounded as follows

$$\mathbb{E}\|\hat{\boldsymbol{g}}^t - \nabla \mathcal{L}(\boldsymbol{w}^t)\|^2 = \mathbb{E}\|\frac{\tilde{h}}{n-f}\bar{\boldsymbol{g}}_{\tilde{\mathcal{H}}}^t + \frac{\tilde{f}}{n-f}\bar{\boldsymbol{g}}_{\tilde{\mathcal{B}}}^t - \nabla \mathcal{L}(\boldsymbol{w}^t)\|^2 \tag{93}$$

$$= \mathbb{E}\|\frac{\tilde{h}}{n-f}(\bar{\boldsymbol{g}}_{\tilde{\mathcal{H}}}^t - \nabla \mathcal{L}(\boldsymbol{w}^t)) + \frac{\tilde{f}}{n-f}(\bar{\boldsymbol{g}}_{\tilde{\mathcal{B}}}^t - \nabla \mathcal{L}(\boldsymbol{w}^t))\|^2 \tag{94}$$

$$\leq \frac{2\tilde{h}^2}{(n-f)^2}\mathbb{E}\|\bar{\boldsymbol{g}}_{\tilde{\mathcal{H}}}^t - \bar{\boldsymbol{g}}^t\|^2 + \frac{2\tilde{f}^2}{(n-f)^2}\mathbb{E}\|\bar{\boldsymbol{g}}_{\tilde{\mathcal{B}}}^t - \bar{\boldsymbol{g}}^t\|^2 \tag{95}$$

We further bound $\mathbb{E}\|\bar{\boldsymbol{g}}_{\tilde{\mathcal{H}}}^t - \bar{\boldsymbol{g}}^t\|^2$ and $\mathbb{E}\|\bar{\boldsymbol{g}}_{\tilde{\mathcal{B}}}^t - \bar{\boldsymbol{g}}^t\|^2$.

First, we bound $\|\sum_{h \in \tilde{\mathcal{H}}} \boldsymbol{g}_h^t - \nabla \mathcal{L}(\boldsymbol{w}^t)\|$ as follows.

$$\mathbb{E}\|\bar{\boldsymbol{g}}_{\tilde{\mathcal{H}}}^t - \nabla \mathcal{L}(\boldsymbol{w}^t)\|^2 = \mathbb{E}\|(\bar{\boldsymbol{g}}_{\tilde{\mathcal{H}}}^t - \frac{1}{\tilde{h}}\sum_{h \in \tilde{\mathcal{H}}} \nabla \mathcal{L}_i(\boldsymbol{w}^t)) + (\frac{1}{\tilde{h}}\sum_{h \in \tilde{\mathcal{H}}} \nabla \mathcal{L}_i(\boldsymbol{w}^t) - \bar{\boldsymbol{g}}^t)\|^2 \tag{96}$$

$$\leq 2\mathbb{E}\|\bar{\boldsymbol{g}}_{\tilde{\mathcal{H}}}^t - \frac{1}{\tilde{h}}\sum_{h \in \tilde{\mathcal{H}}} \nabla \mathcal{L}_i(\boldsymbol{w}^t)\|^2 + 2\mathbb{E}\|\frac{1}{\tilde{h}}\sum_{h \in \tilde{\mathcal{H}}} \nabla \mathcal{L}_i(\boldsymbol{w}^t) - \bar{\boldsymbol{g}}^t\|^2 \tag{97}$$

$$\leq \sigma^2/\tilde{h} + \kappa^2 \tag{98}$$

Then we bound $\mathbb{E}\|\bar{\boldsymbol{g}}_{\tilde{\mathcal{B}}}^t - \bar{\boldsymbol{g}}^t\|^2$. According to Lemma 1, Byzantine gradients away from the optimal gradient will be directly filtered. Therefore, with probability $1 - \varepsilon$

$$\|\boldsymbol{g}_b^t - \nabla \mathcal{L}(\boldsymbol{w}^t)\| \leq \mathcal{O}((\kappa + \sigma)(1 + c + \frac{1}{\sqrt{n-f}})(1 + \sqrt{\frac{n}{p\varepsilon}})), \quad b \in \tilde{\mathcal{B}} \tag{99}$$

$$\tag{100}$$

Then, we have

$$\mathbb{E}\|\bar{\boldsymbol{g}}_{\tilde{\mathcal{B}}}^t - \nabla\mathcal{L}(\boldsymbol{w}^t)\|^2 \leq \mathcal{O}((\kappa^2 + \sigma^2)(1 + c^2 + \frac{1}{n-f})(1 + \frac{n}{p})) := C_1^2 \tag{101}$$

The elimination of $\varepsilon$ is due to the sub-Gaussian property of $\bar{\boldsymbol{g}}_{\tilde{\mathcal{B}}}^t - \nabla\mathcal{L}(\boldsymbol{w}^t)$, which comes from the Gaussian property of benign gradients.

Combine Equations (98) and (101), $\mathbb{E}\|\hat{\boldsymbol{g}}^t - \nabla\mathcal{L}(\boldsymbol{w}^t)\|$ is finally bounded as follows

$$\mathbb{E}\|\hat{\boldsymbol{g}}^t - \nabla\mathcal{L}(\boldsymbol{w}^t)\|^2 \tag{102}$$

$$\leq \frac{2\tilde{h}^2}{(n-f)^2}(\sigma^2/\tilde{h} + \kappa^2) + \frac{2\tilde{f}^2}{(n-f)^2}C_1(1 + 1/p) \tag{103}$$

$$\leq \frac{2(n-2f)^2}{(n-f)^2}(\sigma^2/(n-2f) + \kappa^2) + \frac{2f^2}{(n-f)^2}C_1^2 \tag{104}$$

$$:= C_2 \tag{105}$$

Then, we bound inner product term $\mathbb{E}\langle\nabla\mathcal{L}(\boldsymbol{w}^t), \hat{\boldsymbol{g}}^t - \nabla\mathcal{L}(\boldsymbol{w}^t)\rangle$.

$$|\mathbb{E}\langle\nabla\mathcal{L}(\boldsymbol{w}^t), \hat{\boldsymbol{g}}^t - \nabla\mathcal{L}(\boldsymbol{w}^t)\rangle| \leq \mathbb{E}|\langle\nabla\mathcal{L}(\boldsymbol{w}^t), \hat{\boldsymbol{g}}^t - \nabla\mathcal{L}(\boldsymbol{w}^t)\rangle| \tag{106}$$

$$\leq \mathbb{E}|\langle\nabla\mathcal{L}(\boldsymbol{w}^t), \hat{\boldsymbol{g}}^t - \nabla\mathcal{L}(\boldsymbol{w}^t)\rangle| \tag{107}$$

$$\leq \mathbb{E}\|\langle\nabla\mathcal{L}(\boldsymbol{w}^t)\| \cdot \|\hat{\boldsymbol{g}}^t - \nabla\mathcal{L}(\boldsymbol{w}^t)\| \tag{108}$$

$$\leq \mathbb{E}[\frac{1}{2}\|\langle\nabla\mathcal{L}(\boldsymbol{w}^t)\|^2 + 2\|\hat{\boldsymbol{g}}^t - \nabla\mathcal{L}(\boldsymbol{w}^t)\|^2] \tag{109}$$

$$\leq \frac{1}{2}E\|\langle\nabla\mathcal{L}(\boldsymbol{w}^t)\|^2 + 2C_2 \tag{110}$$

Combine Equations (91), (104) and (110), we have

$$\mathcal{L}(\boldsymbol{w}^t) - \mathcal{L}(\boldsymbol{w}^{t+1}) \geq (\frac{1}{2}\eta - \frac{L}{4}\eta^2)\mathbb{E}\left\|\nabla\mathcal{L}(\boldsymbol{w}^t)\right\|^2 - (\frac{1}{2}\eta - \frac{L}{2}\eta^2)C_2. \tag{111}$$

Sum Equation (90) over $t = 0, 1, \ldots, T-1$ and take expectation, then we have

$$\mathbb{E}[\mathcal{L}(\boldsymbol{w}^0) - \mathcal{L}(\boldsymbol{w}^T)] \geq (\frac{1}{2}\eta - \frac{L}{4}\eta^2)\sum_{t-1}^{T}\mathbb{E}\left\|\nabla\mathcal{L}(\boldsymbol{w}^t)\right\|^2 - T(\frac{1}{2}\eta - \frac{L}{2}\eta^2)C_2. \tag{112}$$

Take $\eta = 1/2L$, and consider that the loss function is generally non-negative, e.g., cross-entropy loss, $\ell_2$ loss,

$$\mathbb{E}\mathcal{L}(\boldsymbol{w}^0) \geq \frac{3}{16L}\sum_{t-1}^{T}(\mathbb{E}\left\|\nabla\mathcal{L}(\boldsymbol{w}^t)\right\|^2 - \frac{2}{3}C_2), \tag{113}$$

which completes the proof. $\square$

## C   COMPARASION AGAINST RECENT WORKS

Recent works (Karimireddy et al., 2022; Allen-Zhu et al., 2020; El-Mhamdi et al., 2021) also analyze the convergence of Byzantine-robust FL in the non-IID setting. We all provide an upper bound on the gradient norms for the convergence analysis. We all admit that convergence in presence of Byzantines may be impossible due to non-IID data, i.e., $\|\nabla\mathcal{L}(\boldsymbol{w})\|$ may never decrease to zero. And the non-IID degree plays a key role in the upper bound. Technically, we improve convergence in different ways. In particular, Allen-Zhu et al. (2020) show how server momentum or history gradients can help convergence. Karimireddy et al. (2022) considers the combined effect of server momentum

and gradient bucketing. El-Mhamdi et al. (2021) considers a decentralized setting and minimizes the upper bound from the point of view of the robust AGR design. Our method considers how gradient decomposition can help convergence. In this sense, our convergence analysis is orthogonal to the above works and may be combined with them to achieve a better upper bound.

## D  EXPERIMENT SETUP

### D.1  SETUP FOR MAIN EXPERIMENTS IN SECTION 7

**Data distribution.** For CIFAR-10, CIFAR-100 (Krizhevsky et al., 2009) and ImageNet-12, we use Dirichlet distribution to generate non-IID data by following Yurochkin et al. (2019); Li et al. (2021). In particular, for each client $i$, we sample $p_i^y \sim \text{Dir}(\beta)$ and allocate a $p_i^y$ proportion of the data of label $y$ to client $i$, where $\text{Dir}(\beta)$ represents the Dirichlet distribution with a concentration parameter $\beta$. We follow Li et al. (2021) and set the number of clients $n = 50$ and the concentration parameter $\beta = 0.5$ as default.

**Other setups.** The setups for datasets FEMNIST (Caldas et al., 2018), CIFAR-10 (Krizhevsky et al., 2009), CIFR-100 (Krizhevsky et al., 2009) and ImageNet-12 (Russakovsky et al., 2015) are listed in below Table 4.

Table 4: Default experimental settings for FEMNIST, CIFAR-10, CIFAR-100 and ImageNet-12.

| Dataset | FEMNIST | CIFAR-10 | CIFAR-100 | ImageNet-12 |
|---|---|---|---|---|
| Architecture | CNN (Caldas et al., 2018) | AlexNet (Krizhevsky et al., 2017) | SqueezeNet (Iandola et al., 2016) | ResNet-18 (He et al., 2016) |
| # Communication rounds | 1000 | 200 | 400 | 200 |
| Client sample ratio | 0.005 | 0.1 | 0.1 | 0.1 |
| # Local epochs | 1 | 5 | 1 | 1 |
| Optimizer | SGD | SGD | SGD | SGD |
| Batch size | 64 | 64 | 64 | 128 |
| Learning rate | 0.5 | 0.1 | 0.1 | 0.1 |
| Momentum | 0.5 | 0.5 | 0.5 | 0.9 |
| Weight decay | 0.0001 | 0.0001 | 0.0001 | 0.0001 |
| Learning rate decay | No | No | No | Reduce to 0.01 after 100-th communication round |
| Gradient clipping | Yes | Yes | Yes | Yes |
| Clipping norm | 2 | 2 | 2 | 2 |

Three types of attacks based on the adversary's knowledge are considered:

- Agnostic attack: the adversary knows neither honest gradients nor the AGR.

- Partial knowledge attack: the adversary only has the knowledge of honest gradients.

- Omniscient attack: the adversary knows both honest gradients and the AGR.

Among the six attacks considered: BitFlip (Allen-Zhu et al., 2020), LabelFlip (Allen-Zhu et al., 2020) are agnostic attacks; LIE (Baruch et al., 2019), Min-Max (Shejwalkar & Houmansadr, 2021), Min-Sum (Shejwalkar & Houmansadr, 2021) are partial knowledge attacks; and IPM (Xie et al., 2020) is an omniscient attack.

The hyperparameters of six attacks: BitFlip (Allen-Zhu et al., 2020), LabelFlip (Allen-Zhu et al., 2020), LIE (Baruch et al., 2019), Min-Max (Shejwalkar & Houmansadr, 2021), Min-Sum (Shejwalkar & Houmansadr, 2021), IPM (Xie et al., 2020), are listed in below Table 5.

The hyperparameters of six defenses: Multi-Krum (Blanchard et al., 2017), Bulyan (Guerraoui et al., 2018), Median (Yin et al., 2018), RFA (Pillutla et al., 2019), DnC (Shejwalkar & Houmansadr, 2021), RBTM (El-Mhamdi et al., 2021), are listed in below Table 5.

Table 5: The hyperparameters of six attacks.

| Attacks | Hyperparameters |
|---------|-----------------|
| BitFlip | N/A |
| LabelFlip | N/A |
| LIE | $z = 1.5$ |
| Min-Max | $\gamma_{\text{init}} = 10, \tau = 1 \times 10^{-5}, \boldsymbol{\delta}$: coordinate-wise standard deviation |
| Min-Sum | $\gamma_{\text{init}} = 10, \tau = 1 \times 10^{-5}, \boldsymbol{\delta}$: coordinate-wise standard deviation |
| IPM | # eval $= 2$ |

Table 6: The default hyperparameters of the AGRs.

| AGRs | Hyperparameters |
|------|-----------------|
| Multi-Krum | N/A |
| Bulyan | N/A |
| Median | N/A |
| RFA | $T = 3$ |
| DnC | $c = 4, \text{niters} = 1, b = 10000$ |
| RBTM | N/A |

### D.2 SETUP FOR EXPERIMENTS ON THE NUMBER OF SUB-VECTORS IN SECTION 7

The number of client is set to $n = 50$. The samples are partitioned in a Dirichlet manner with concentration parameter $\beta = 0.5$. Please refer to Sec. 7.1 for the details of Dirichlet partition. The number of Byzantine clients is set to $f = 10$. LIE (Baruch et al., 2019) attack with $z = 1.5$ considered.

We use AlexNet (Krizhevsky et al., 2017) as the model architecture. The number of communication rounds is set to 500. In each communication round, all client participate in the training.

For local training, the number of local epochs is set to 1, batch size is set to 64, the optimizer is set to SGD. For SGD optimizer, learning rate is set to 0.1, momentum is set to 0.5, weight decay coeffecient is set to 0.0001. We also adopt gradient clipping with clipping norm 2.

Two defenses are considered: a radical AGR Multi-Krum (Blanchard et al., 2017) and a conservative AGR Bulyan (Guerraoui et al., 2018).

## E  GAIN MITIGATES THE DEVIATION OF AGGREGATED GRADIENTS

In Sec. 6, we claim that our GAIN method can reduce the deviation of aggregated gradient $\hat{\boldsymbol{g}}$ from the average of honest gradients $\boldsymbol{g}$. To verify this fact, we compare the deviation of the aggregated gradient of different defenses and their GAIN variants in Figure 3. In particular, we use $\|\hat{\boldsymbol{g}} - \boldsymbol{g}\|$, the distance between the aggregated gradient $\hat{\boldsymbol{g}}$ and the average of honest gradients $\boldsymbol{g}$ to measure the deviation degree. As shown in Figure 3, the gradient deviation degree of GAIN-enhanced defenses is much lower than their original versions as expected, which validates that our GAIN can mitigate the gradient deviation.

## F  RESULTS ON DIFFERENT NUMBER OF CLIENTS.

We also conduct experiments across different number of clients. Table 7 demonstrates the results of different defenses under LIE attack across $n = \{75, 100\}$ clients on CIFAR-10 dataset. Note that the number of Byzantine clients is set to $f = 0.2 \cdot n$ correspondingly. Other setups follow the default

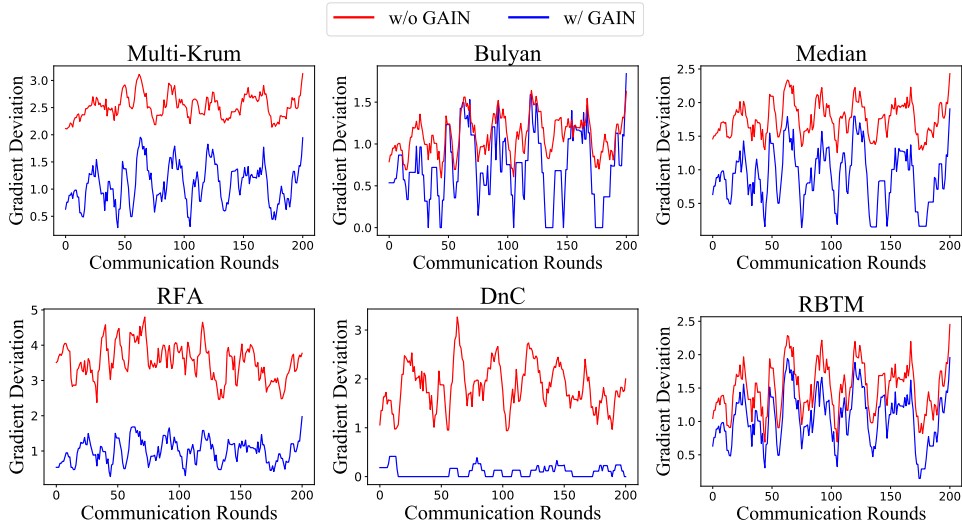

Figure 3: The gradient deviation $\|\hat{g} - g\|$ of six different defenses w/ and w/o GAIN under LIE attack on CIFAR-10. The lower the better.

setup of the main experiments in Sec. 3 and Appendix D. As evidenced by Table 7, integrating current robust AGRs into our GAIN outperforms their original versions across all client numbers.

Table 7: Accuracy (mean±std) of different defenses against LIE attack under different client numbers on CIFAR-10.

| $n$ | Multi-Krum | Multi-Krum+GAIN | Bulyan | Bulyan+GAIN | Median | Median+GAIN |
|---|---|---|---|---|---|---|
| 75 | $28.72 \pm 0.71$ | $\mathbf{54.89} \pm 0.16$ | $23.37 \pm 1.22$ | $\mathbf{51.11} \pm 0.00$ | $44.89 \pm 2.98$ | $\mathbf{52.22} \pm 1.64$ |
| 100 | $32.49 \pm 1.22$ | $\mathbf{56.51} \pm 0.01$ | $21.93 \pm 0.55$ | $\mathbf{46.49} \pm 1.33$ | $33.82 \pm 0.21$ | $\mathbf{46.12} \pm 0.17$ |

| $n$ | RFA | RFA+GAIN | DnC | DnC+GAIN | RBTM | RBTM+GAIN |
|---|---|---|---|---|---|---|
| 75 | $16.89 \pm 1.38$ | $\mathbf{49.85} \pm 0.06$ | $59.31 \pm 1.33$ | $\mathbf{59.75} \pm 0.42$ | $45.06 \pm 0.96$ | $\mathbf{50.24} \pm 0.31$ |
| 100 | $14.01 \pm 1.34$ | $\mathbf{49.85} \pm 1.97$ | $58.88 \pm 1.45$ | $\mathbf{59.61} \pm 1.19$ | $40.38 \pm 0.48$ | $\mathbf{47.02} \pm 0.03$ |

