# OpenReview forum: "GAIN: Enhancing Byzantine Robustness in Federated Learning with Gradient Decomposition"
_ICLR.cc/2023/Conference — Submitted to ICLR 2023_

### Official Review · Reviewer_TG9e · 2022-10-23

**Confidence:** 4
**Correctness:** 4
**Technical Novelty And Significance:** 3
**Empirical Novelty And Significance:** 3
**Recommendation:** 8

**Clarity, Quality, Novelty And Reproducibility:**

Code is provided to reproduce all experiments. The idea of gradient decomposition is novel. The writing is clear and easy to follow.

**Strength And Weaknesses:**

Strength

1. This paper tries to tackle Byzantine attacks in FL under non-IID setting. The topic is interesting and useful since data distribution is mostly non-IID in the real world.
2. The authors point out the root reasons from a new lens, in order to explain why the existing aggregation rules fail under the non-IID setting. Then they propose a novel gradient decomposition scheme (GAIN) to mitigate the gradient deviation and improve the performance. The method is novel and easy to follow.
3. Empirical results are sound. Extensive experiments under three different types of attacks can well support the effectiveness of GAIN.

Weaknesses

1. There are various types of non-IID. Does the proposed method apply to all these types of non-IID settings? More discussion is expected.
2. In the proof for Proposition 1: the illustration for inequality (32) is confusing, more explanation is expected; the illustration for inequality (33) is missing.
3. What is $\delta_b$ in equation (51)?


**Summary Of The Paper:**

This paper provides a comprehensive study on the Byzantine robustness of federated learning under the non-IID setting. The authors show that existing robust aggregation rules fail to defend against Byzantine attacks under the non-IID setting, and explain the root reasons by proposing two novel concepts: identification failure and integrity failure. Then, the authors propose a novel gradient decomposition scheme (GAIN) that can address both failures. Under standard assumptions, the authors derive that GAIN can mitigate the deviation of aggregated gradient, thus improving the performance. Experiments show the benefits of the proposed GAIN.

**Summary Of The Review:**

While there are some weaknesses, I think this work is above the threshold given the novelty and sound results in tackling the challenge of Byzantine robustness under the non-IID setting.

---

> ### Author Response · Authors · 2022-11-12
> **Author Response**
>
> We thank the reviewer for taking the time to review our paper carefully and verify our theoretical results.
>
> ---
>
> **Q1: There are various types of non-IID. Does the proposed method apply to all these types of non-IID settings? More discussion is expected.**
>
> **R1:**
>
> Thanks for the thoughtful comment.
>
> The proposed method can be applied to general non-IID types, including covariate shift, prior probability shift, concept shift, etc [1].
> In experiments, we follow previous works [2, 3] and validate the efficacy of the proposed method in a synthetic prior probability shift setting (CIFAR-10, ImageNet-12, CIFAR-100) and a real-world concept shift setting (FEMNIST).
>
> ---
>
> **Q2: In the proof for Proposition 1: the illustration for inequality (32) is confusing, more explanation is expected; the illustration for inequality (33) is missing.**
>
> **R2:**
>
> Thank you for pointing this out.
>
> Inequality (32) comes from Cauchy inequality, and inequality (33) is a combined result of Equation (28), Assumption 2, and Assumption 3.
>
> We have clarified it in our revision. Please find more details in Appendix B.
>
> ---
>
> **Q3: What is $\delta_b$ in equation (51)?**
>
> **R3:**
> Thank you for pointing out the typo.
>
> Sorry for missing this important definition. In fact, $\delta_b = \mathbb{E} \Vert \boldsymbol{g}_{b}^{(q)} - \bar{\boldsymbol{g}}^{(q)} \Vert$.
>
> We have specified the definition in the revised paper. Please find it in Appendix B.
>
> ---
>
> We hope the clarifications above could improve your opinion of our paper.
>
> ---
>
> **References:**
>
> [1] Kairouz, Peter, et al. "Advances and open problems in federated learning." Foundations and Trends® in Machine Learning 14.1–2 (2021): 1-210.
>
> [2] Tang Z., Zhang Y., et al. " Virtual Homogeneity Learning: Defending against Data Heterogeneity in Federated Learning." International Conference on Machine Learning. PMLR, 2022.
>
> [3] Luo, Mi, et al. "No fear of heterogeneity: Classifier calibration for federated learning with non-iid data." Advances in Neural Information Processing Systems 34 (2021): 5972-5984.

---

> > ### Comment · Reviewer_TG9e · 2022-11-16
> > **Concerns have been addressed**
> >
> > The reviewer thanks the responses from the authors. Most of my concerns have been addressed, therefore, I would like to raise my original score.

---

### Official Review · Reviewer_Fwja · 2022-10-23

**Confidence:** 4
**Correctness:** 4
**Technical Novelty And Significance:** 4
**Empirical Novelty And Significance:** 4
**Recommendation:** 6

**Clarity, Quality, Novelty And Reproducibility:**

The paper is novel and technically sound with adequate reproducibility. The structure of the paper can be further polished.

**Strength And Weaknesses:**

Strength:

•	The investigated problem, byzantine-robust FL over non-IID data, is important. The failure mechanism of existing aggregation rules over non-IID data is well investigated.

•	The proposed method is novel and can adequately address the failures of existing aggregation rules over non-IID data.

•	The authors theoretically justify the efficacy of the proposed gradient decomposition technique.

Weaknesses:

•	Seems that Eq. (8) should belong to the identification phase?

•	Please explain why computing identification score is put in the aggregation phase. It would be better to reorganize Sec. 5.

•	The paper mentioned "We consider six representative attacks covering the three attack types described in Sec. 3". It would be clearer to explicitly define three attack types in Sec. 3.

•	It would be better if the authors can test on 1-2 additional larger datasets, e.g., cifar100. This could make the results more convincing.


**Summary Of The Paper:**

This paper focuses on byzantine robustness of FL. Typically, the server uses robust aggregation rules to ensure that byzantine clients do not hinder learning. However, the performance of most aggregation rules is degraded when data is non-IID across different clients. The authors reveal the root causes of the performance degradation -- identification failure and integrity failure. Then they propose a novel gradient decomposition scheme that can address the above failures and adapt existing aggregation rules to non-IID data. Theoretical analysis and numerical experiments validate the efficacy of the proposed method.

**Summary Of The Review:**

This paper provides an interesting insight on byzantine-robust FL over non-IID data. The proposed gradient decomposition method is well motivated. The authors also theoretically analyze the efficacy of the proposed method. Further experiments on larger datasets are expected. In general, this paper has publication merits, but some issues need to be addressed.

---

> ### Author Response · Authors · 2022-11-12
> **Author Response**
>
> Thank you for taking the time to review our paper and give suggestions to improve clarity. We would like to address the concerns as follows.
>
> ---
>
> **Q1 & 2: Seems that Eq. (8) should belong to the identification phase? Please explain why computing identification score is put in the aggregation phase. It would be better to reorganize Sec. 5.**
>
> **R1 & 2:**
>
> Thank you for the kind suggestion.
>
> We have followed your suggestion and reorganized Sec. 5. In particular, we move both Eq. (8) (apply robust AGR to decomposed sub-vectors) and Eq. (10) (compute identification score) to the identification phase. Please kindly refer to Sec. 5 for more details.
>
> ---
>
> **Q3: The paper mentioned "We consider six representative attacks covering the three attack types described in Sec. 3". It would be clearer to explicitly define three attack types in Sec. 3.**
>
> **R3:**
>
> Thanks for pointing this out. Three types of attacks based on the adversary’s knowledge are considered:
> - Agnostic attack: the adversary knows neither honest gradients nor the AGR.
> - Partial knowledge attack: the adversary only has the knowledge of honest gradients.
> - Omniscient attack: the adversary knows both honest gradients and the AGR.
>
> Among the six representative attacks considered: BitFlip, LabelFlip are agnostic attacks; LIE, Min-Max, Min-Sum are partial knowledge attacks; and IPM is an omniscient attack.
>
> We have added the discussion about attack types in Appendix C. Please kindly find it in the revision for more details.
>
> ---
>
> **Q4: It would be better if the authors can test on 1-2 additional larger datasets, e.g., cifar100. This could make the results more convincing.**
>
> **R4:**
>
> Thanks for your suggestion.
>
> Following your suggestion, we have run experiments on CIFAR-100. The experimental results are listed in Table 1 below. These results
> indicate that integrating current defenses into our GAIN method can achieve better performance than their original versions on CIFAR-100. (Please refer to Table 1 in the revision for more details.)
>
> Table 1: Accuracy (mean $\pm$ std) of different defenses under 6 attacks on CIFAR-100.
>
> | Attack | BitFlip | LabelFlip | LIE | Min-Max | Min-Sum | IPM |
> |:-:|-:|-:|-:|-:|-:|-:|
> | Multi-Krum | 34.27 $\pm$ 0.28 | 35.57 $\pm$ 0.94 | 17.17 $\pm$ 0.08 | 16.77 $\pm$ 0.78 | 22.89 $\pm$ 0.61 | 15.93 $\pm$ 2.00 |
> | Multi-Krum+GAIN | **42.41** $\pm$ 0.58 | **42.55** $\pm$ 0.12 | **27.81** $\pm$ 0.32 | **31.18** $\pm$ 1.48 | **41.33** $\pm$ 0.50 | **42.62** $\pm$ 1.53 |
> ||
> | Bulyan | 35.77 $\pm$ 0.18 | 42.60 $\pm$ 0.07 | 35.41 $\pm$ 0.40 | 35.53 $\pm$ 1.38 | 39.13 $\pm$ 0.12 | 40.27 $\pm$ 1.64 |
> | Bulyan+GAIN | **42.28** $\pm$ 1.61 | **43.77** $\pm$ 0.46 | **38.39** $\pm$ 0.19 | **36.33** $\pm$ 1.51 | **40.73** $\pm$ 0.39 | **42.88** $\pm$ 0.14 |
> ||
> | Median | 36.62 $\pm$ 0.12 | 41.64 $\pm$ 0.76 | 22.75 $\pm$ 0.04 | 23.21 $\pm$ 0.71 | 30.68 $\pm$ 0.26 | 40.98 $\pm$ 0.38 |
> | Median+GAIN | **42.41** $\pm$ 0.66 | **42.62** $\pm$ 0.09 | **35.16** $\pm$ 1.08 | **36.46** $\pm$ 0.10 | **41.08** $\pm$ 0.04 | **43.63** $\pm$ 2.85 |
> ||
> | RFA | 21.32 $\pm$ 0.84 | 28.76 $\pm$ 1.33 | 25.63 $\pm$ 0.20 | 26.46 $\pm$ 1.83 | 28.33 $\pm$ 0.93 | 21.36 $\pm$ 0.54 |
> | RFA+GAIN | **42.64** $\pm$ 0.44 | **42.42** $\pm$ 0.25 | **26.30** $\pm$ 1.08 | **30.30** $\pm$ 0.12 | **41.09** $\pm$ 0.66 | **43.45** $\pm$ 0.52 |
> ||
> | DnC | 41.77 $\pm$ 0.62 | 42.93 $\pm$ 0.07 | 42.95 $\pm$ 1.03 | 40.15 $\pm$ 0.70 | 40.02 $\pm$ 1.07 | 41.23 $\pm$ 2.29 |
> | DnC+GAIN | **43.35** $\pm$ 0.41 | **43.57** $\pm$ 1.11 | **43.64** $\pm$ 0.11 | **41.66** $\pm$ 0.78 | **41.02** $\pm$ 1.39 | **43.25** $\pm$ 0.43 |
> ||
> | RBTM | 36.35 $\pm$ 0.17 | 42.67 $\pm$ 1.55 | 24.06 $\pm$ 0.09 | 26.24 $\pm$ 1.04 | 36.51 $\pm$ 0.40 | 43.12 $\pm$ 1.12 |
> | RBTM+GAIN | **43.44** $\pm$ 0.81 | **43.19** $\pm$ 2.65 | **33.14** $\pm$ 0.58 | **34.35** $\pm$ 0.76 | **41.51** $\pm$ 0.93 | **43.20** $\pm$ 0.76 |
>
> We hope the new results can address your concerns about the performance of the proposed method on larger datasets.
>
> ---
>
> We hope our responses above can adequately address your concerns.

---

> > ### Comment · Reviewer_Fwja · 2022-12-01
> > **post-response**
> >
> > Thank you for the response, I'm satisfied with newly added experiments. I keep positive for this submission.

---

### Official Review · Reviewer_Bimn · 2022-10-23

**Confidence:** 5
**Clarity, Quality, Novelty And Reproducibility:** Maybe it is not novel.
**Correctness:** 3
**Technical Novelty And Significance:** 1
**Empirical Novelty And Significance:** 1
**Recommendation:** 3

**Strength And Weaknesses:**

Weaknesses:
1. The writing is not clear, where partition of set {1,...,d}, which may be very critical?

2. The word "decomposition" may not fit.

3. The method is very simple. They stack other's method except set partition?



**Summary Of The Paper:**

In this paper, the authors try to solve the identification failure and integrity failure in federated learning for non-iid setting. In order to address
both failures, they propose GAIN, a gradient decomposition scheme that can help adapt existing robust algorithms to heterogeneous datasets. In theory, they show that integrating exisiting robust AGRs into GAIN can mitigate the deviation of aggregated gradient, thus improve the performance. Experiments on various real-world datasets verify the efficacy of proposed GAIN.

**Summary Of The Review:**

Maybe it is not suitable for this top conference

---

> ### Author Response · Authors · 2022-11-12
> **Author Response**
>
> Thanks very much for your time reviewing our paper and the thoughtful questions. Following your suggestions, we have revised our paper. We hope the new version and the response below can adequately address your concerns.
>
> ---
> **Q1: The writing is not clear, where partition of set {$ \{ 1,...,d \} $}, which may be very critical?**
>
> **R1:**
>
> Thanks for pointing this out. We are sorry for confusing the reviewer.
>
> In fact, the partition of set {$ \{ 1,...,d \} $} in our method can be arbitrary as long as it satisfies Eq. (6). Particularly, we randomly partition set {$\{1,...,d\}$} into $p$ equally-sized subsets in our experiments. And the gradients are decomposed according to the partition. We have specified it in the revised version of our paper in Sec. 7.1.
>
> ---
> **Q2: The word "decomposition" may not fit.**
>
> **R2:**
>
> Thank you for your comment.
>
> We would like to further clarify that “decomposition” indicates the process of *decomposing each gradient into sub-vectors* for further gradient identification.
>
> As shown in Sec. 6, decomposing gradients helps to mitigate the deviation of aggregated gradient, thus improving the performance of the global model. Therefore, we consider the decomposition process the core part of our method and name our method gradient decomposition.
>
> ---
> **Q3: The method is very simple. They stack other's method except set partition?**
>
> **R3:**
>
> Thanks for your question.
>
> We agree that our gradient decomposition method needs to be combined with other robust aggregation rules to defend against Byzantine attacks. However, we would like to argue that our gradient decomposition method plays a key role in defending against Byzantine attacks in the **non-IID** setting. As discussed in Sec. 4, the performance of most existing robust aggregation rules is degraded when the data is non-IID due to the integrity failure and identification failure. Although our method is simple, it can effectively address both failures and adapt these aggregation rules to the non-IID setting.
>
> Besides, both theoretical and empirical evidence are provided to validate that the proposed method is simple but effective. In Sec. 6, we theoretically show the proposed method can mitigate the deviation of the aggregated gradient from the average of honest gradients in the non-IID setting via Proposition 1. In Sec. 7, we empirically validate that combining our method with existing aggregation rules can significantly improve the performance of global models.
>
> Overall, we believe that our proposed method is of significant interest to the FL community. We hope we have cleared your concern. We believe it’s not a good reason to reject a paper because of the simplicity of the method. If a simple method can work well enough, why should we turn to a complex and expensive method?
>
> ---
> We hope above clarifications could help improve your opinion of our paper.

---

> > ### Comment · Reviewer_Bimn · 2022-11-16
> > **Method simple**
> >
> > At the first beginning of your paper, you show that some methods are good for iid case, but fail in non-iid case. So your object is to solve this problem. In classical random forest method, they use the same idea. At the same time, there is one paper about metamask in nips 2022, I am not sure it is yours.

---

> > > ### Author Response · Authors · 2022-11-17
> > > **Our method is different from the mentioned methods**
> > >
> > > We hope the following answers can address your concerns.
> > >
> > > ---
> > >
> > > **Q1:** At the first beginning of your paper, you show that some methods are good for iid case, but fail in non-iid case. So your object is to solve this problem. In classical random forest method, they use the same idea.
> > >
> > > **A1:**
> > >
> > > Sorry we cannot agree with your opinion. Our method is completely different from Random Forest, which can be reflected by the follow-up points:
> > >
> > > - **Solve Non-IID:** Random Forest is designed for centralized learning and has nothing to do with the non-IID challenge in federated learning, while our method considers Byzantine robustness in the existence of non-IID data.
> > >
> > > - **Different Research Topics:** Random Forest is usually designed for classification or regression. By contrast, our method aims to defend against Byzantine gradients in federated learning.
> > >
> > > - **Different Technical implementation:** A random forest consists of an ensemble of tree-structured classifiers depending on a collection of random variables [1, 2]. By contrast, our method does not involve any ensemble technique or tree structure.
> > >
> > > Therefore, our paper is completely different from Random Forest in terms of both research topic and technical implementation.
> > >
> > > ---
> > >
> > > **Q2:** At the same time, there is one paper about metamask in nips 2022, I am not sure it is yours.
> > >
> > > **A2:**
> > >
> > > In terms of MetaMask [3], we want to state the key differences as follows:
> > >
> > > - **Different Research Topics:** [3] studies self-supervised learning and aims to learn discriminative representations from the input data without relying on human annotations. By contrast, we consider the Byzantine robustness in federated learning.
> > >
> > > - **Different Technical implementation:** [3] proposes to assign different weights to different dimensions of representations to adjust their gradient contribution to improve a specific self-supervised task during optimization. The weights are dynamically adjusted in a meta-learning manner. However, our method neither includes the reweighting scheme nor employs a meta-learning paradigm.
> > >
> > > In principle, our work and MetaMask [3] consider totally different topics. Therefore, there are no similarities between [3] and our paper.
> > >
> > > ---
> > >
> > > **References:**
> > >
> > > [1] Breiman, Leo. "Random forests." Machine learning 45.1 (2001): 5-32.
> > >
> > > [2] Cutler, Adele, D. Richard Cutler, and John R. Stevens. "Random forests." Ensemble machine learning. Springer, Boston, MA, 2012. 157-175.
> > >
> > > [3] Li, J., Qiang, W., Zhang, Y., Mo, W., Zheng, C., Su, B., & Xiong, H. " MetaMask: Revisiting Dimensional Confounder for Self-Supervised Learning." Advances in Neural Information Processing Systems 35 (2022).

---

### Official Review · Reviewer_bm6A · 2022-10-25

**Confidence:** 4
**Correctness:** 4
**Technical Novelty And Significance:** 4
**Empirical Novelty And Significance:** 3
**Recommendation:** 8

**Clarity, Quality, Novelty And Reproducibility:**

The quality, clarity and originality of the work are good.

Clarity: The majority of the paper is written clearly; however, some parts require more elaboration; see the weakness section.
Quality: Exploiting the root causes of current robust AGR performance degradation in non-IID settings is an intriguing and important problem. This paper did an excellent job of summarising and explaining why current robust AGRs fail. The proposed gradient decomposition scheme works well, and the paper as a whole is excellent.
Novelty: To my knowledge, the proposed gradient decomposition scheme (GAIN) is novel.
Replicability: The provided code is clear and simple to replicate.


**Strength And Weaknesses:**

Strength:

1) This paper begins a pilot study on the root causes of current robust AGR performance degradation in non-IID settings by proposing two new concepts: integrity failure and identification failure. To the best of my knowledge, this is a groundbreaking study.
2) Following that, the authors propose a novel gradient decomposition scheme (GAIN) to reduce gradient deviation and improve performance. The method is novel and observationally supported.
3) The entire paper is simple to understand. The authors thoroughly review all related work and summarise its benefits and drawbacks.
4) The gradient decomposition technique proposed here is simple but effective, and it works well with existing methods.


Weaknesses:

1) Which of the six defences discussed in Section 7 is conservative and which is radical? GAIN appears to be more appropriate for conservative methods. This phenomenon can be explained further in more detail.
2) According to the paper, the proposed method can reduce the deviation of the aggregated gradient. It is better to provide experiments to prove this.

**Summary Of The Paper:**

In this paper, the authors discuss Byzantine robustness in federated learning when the data is non-iid. In the non-iid setting, the gradient decomposition scheme (GAIN) is used to address the identification and integrity failures of existing defences. GAIN can be combined with a variety of robust aggregation rules to help them adapt to heterogeneous datasets. The authors theoretically demonstrate that GAIN can reduce aggregated gradient deviation, thereby improving performance. The authors provide sufficient experimental results to support GAIN's superiority.

**Summary Of The Review:**

This paper begins a pilot investigation into the root causes of current robust AGR performance degradation in non-IID settings by proposing two new concepts: integrity failure and identification failure. The authors then propose a novel way to defend against Byzantines in FL. Sufficient theoretical analysis and extensive experiments can well support their proposed method's superiority. I believe that this paper will have an impact on the community.

---

> ### Author Response · Authors · 2022-11-12
> **Author Response**
>
> Thank you very much for the positive feedback and valuable comments. We would like to address the concerns as follows:
>
> ---
> **Q1: Which of the six defenses discussed in Section 7 is conservative and which is radical? GAIN appears to be more appropriate for conservative methods. This phenomenon can be explained further in more detail.**
>
> **R1:**
>
> Thanks for the thoughtful question.
>
> Among the six defenses, Bulyan, Median, and RBTM are conservative, and Multi-Krum, RFA, and DnC are radical. We have specified it in our revision. Please kindly refer to paragraph “Baselines” in Sec. 7 for more details.
>
> In the non-IID setting, the gradients are heterogeneous. Excluding honest gradients deviates the aggregated gradients from the average of honest gradients, thus degrading the performance of conservative methods. When the non-IID degree increases, the gradient heterogeneity increases. As a result, the impact of excluding honest gradients may even be larger than incorporating Byzantine gradients. Therefore, the improvement on conservative methods is greater. We have added above discussion to the revised paper. Please find more details in paragraph “Main results”, item (4) in Sec 7.2.
>
> ---
> **Q2: According to the paper, the proposed method can reduce the deviation of the aggregated gradient. It is better to provide experiments to prove this.**
>
> **R2:**
>
> Thank you for the suggestion.
>
> We have run the experiments as suggested and added the results to Figure 3 in Appendix D. In particular, we use $\|\hat{\boldsymbol{g}}-\boldsymbol{g}\|$, the distance between the aggregated gradient $\hat{\boldsymbol{g}}$ and the average of honest gradients $\boldsymbol{g}$ to measure the deviation degree, and compare the gradient deviation of six different defenses w/ and w/o GAIN under LIE attack on CIFAR-10. These results empirically verify that our GAIN can reduce the deviation of the aggregated gradient. Please kindly find it in the revision.
>
> ---
> We hope the above clarifications and new empirical results can help address your concerns.

---

### Comment · Area_Chair_GocS · 2022-11-17
**Relation to recent works on Byzantine robustness in the non-IID case**

Dear Authors

Could you please clarify some items which to me were unclear now even after the first reviews:
- *Convergence guarantees*: Your guarantee requires that the bad gradients g_b (or can I say Byzantine gradients) are strongly separated from the good ones (bottom of page 6). This in my impression can not be called Byzantine robustness then, as you are forcing those clients b to always respect to return gradients sufficiently far away from g. So they need to adhere to a protocol.
- In the paper please reformulate the following aspect: The well-known LIE attack (Baruch et al. 2019) and others have proven already that gradients close to the optimal gradient per round are *NOT enough* to achieve provable robust training over the entire training process. However the current version of the paper here seems to convey the opposite "convergence is not enough (Guerraoui et al., 2018)". These attacks have broken most existing aggregation rules you cite in the paper, even in the IID case. You currently don't discuss how this has been addressed in the literature of recent years by provable Byzantine robust convergence rates (El-Mhamdi et al. 2021, Allen-Zhu et al. 2021, Karimireddy et al. 2022), neither in the IID or non-IID case.
- Could you clarify or give a reference how the earlier aggregation rules (page 6) are *c-resilient AGR*?
- Could you comment on how your *convergence theory* compares to recent robust non-IID convergence results such as (Karimireddy et al. 2022), (El-Mhamdi et al. 2021), as well as this related additional reference [1]. Also the lower bounds of impossibility of learning in the non-IID case even in low-dimensions?
- Could you include such recent aggregators specifically for non-IID also in your *experiments* as baselines, such as (Karimireddy et al. 2022, El-Mhamdi et al. 2021), and [1]?

*Additional References:*

[1] Byzantine-Robust Variance-Reduced Federated Learning over Distributed Non-i.i.d. Data
Jie Peng, Zhaoxian Wu, Qing Ling, Tianyi Chen   https://arxiv.org/abs/2009.08161

---

> ### Author Response · Authors · 2022-11-21
> **New theoretical and empirical results**
>
> Dear Area Chair GocS,
>
> Thank you for your valuable comments. We hope the responses below can address your concerns.

---

> > ### Author Response · Authors · 2022-11-21
> > **Response to Q1**
> >
> > **Q1:** *Convergence guarantees:* Your guarantee requires that the bad gradients g_b (or can I say Byzantine gradients) are strongly separated from the good ones (bottom of page 6). This in my impression can not be called Byzantine robustness then, as you are forcing those clients b to always respect to return gradients sufficiently far away from g. So they need to adhere to a protocol.
> >
> > **A1**:
> > We agree that such an assumption is too strong and impractical for Byzantine robust learning. In the revision, we extend the previous theoretical result to a new convergence result that does NOT rely on this strong assumption.
> >
> > In particular, we provide an upper bound for the sum of gradient norms under standard assumptions.
> >
> > **Proposition 1** Suppose Assumptions 1 to 4 hold, and let $\eta=1/2L$. Given a $c$-resilient robust AGR $A $, we start from $w^0$ and run GAIN for $T$ communication rounds, it satisfies
> > $$\mathcal{L}(w^0)\ge\frac{3}{16L}\sum_{t=1}^T(\Vert\nabla\mathcal{L}(w^t)\Vert^2-e^2),$$
> > where
> > $$e^2=\mathcal{O}(\frac{f^2}{(n-f)^2}(\kappa^2+\sigma^2)(1+c^2+\frac{1}{n-f})(1+\frac{n}{p})).$$
> >
> > In Proposition 1, error term $e^2$ is a constant, and $\mathcal{L}(w^0)$ is the initial loss. As the number of communication round $T$ increases, GAIN is bound to find an approximate optimal $w$ with $\Vert\nabla\mathcal{L}(w)\Vert\le e$.
> >
> > Please kindly find more details in Sec. 6. And please check **A4** for comparisons between our convergence analysis and existing works.

---

> > ### Author Response · Authors · 2022-11-21
> > **Response to Q2**
> >
> > **Q2:** In the paper please reformulate the following aspect: The well-known LIE attack (Baruch et al. 2019) and others have proven already that gradients close to the optimal gradient per round are NOT enough to achieve provable robust training over the entire training process. However the current version of the paper here seems to convey the opposite "convergence is not enough (Guerraoui et al., 2018)". These attacks have broken most existing aggregation rules you cite in the paper, even in the IID case. You currently don't discuss how this has been addressed in the literature of recent years by provable Byzantine robust convergence rates (El-Mhamdi et al. 2021, Allen-Zhu et al. 2021, Karimireddy et al. 2022), neither in the IID or non-IID case.
> >
> > **A2:**
> >
> > **Contradictions with existing works:** We agree that consistent small perturbations of gradients across communication rounds may hurt training. And the previous theoretical guarantee of our method is insufficient, which may confuse the readers. In our revision, we develop a new convergence result based on the previous result, which can avoid such misunderstandings. Please check our convergence analysis in Sec. 6 in the revision.
> >
> > **Discussion of recent works:** The convergence results of the mentioned works (El-Mhamdi et al. 2021, Allen-Zhu et al. 2021, Karimireddy et al. 2022) all admit that it is impossible to completely get rid of the impact of Byzantines due to the non-IIDness and the variance that comes from randomness of SGD, i.e., a classical convergence is impossible. However, they all provide similar weaker guarantees, i.e., after a certain number of communication rounds, their algorithms are guaranteed to reach an approximate optimal point. Please refer to **A4** for more comparisons between our results and the existing ones.

---

> > > ### Comment · Area_Chair_GocS · 2022-11-24
> > > **Question on convergence result correctness, and implications in the IID special case**
> > >
> > > In the newly added proof for the completely changed main Proposition, a key Lemma 1 (on page 14) still assumes that Byzantine gradients must be clearly separated from regular ones. Could you clarify how the new proof works, and if you will be able to remove this assumption or not?
> > >
> > > thanks

---

> > > > ### Author Response · Authors · 2022-11-26
> > > > **Author Response**
> > > >
> > > > Dear Area Chair GocS,
> > > > Thank you for your valuable time to check the response. We hope the new answers below can address your concerns.
> > > >
> > > > ---
> > > >
> > > > **Q1:** In the newly added proof for the completely changed main Proposition, a key Lemma 1 (on page 14) still assumes that Byzantine gradients must be clearly separated from regular ones. Could you clarify how the new proof works, and if you will be able to remove this assumption or not?
> > > >
> > > > **A1:**
> > > > We want to clarify that the Byzantine separation is not an assumption but a particular case in our casework.
> > > >
> > > > More specifically, in our proof of new Proposition 1, we divide the behavior of the Byzantine gradients into the following two cases:
> > > > - Case 1: Byzantine gradients are separated from the regular ones and can be filtered by our GAIN method.
> > > > - Case 2: Byzantine gradients are close enough to the regular ones and can circumvent the defense.
> > > >
> > > > Lemma 1 specifies when the Byzantine gradients can be filtered (Case 1). And for Case 2, the impact of Byzantine gradients on the aggregated gradients is limited.
> > > > In this way, we consider both cases and derive a unified upper bound on the deviation of the aggregated gradients. Then we can use the upper bound to obtain a convergence result of our GAIN method.
> > > >
> > > > We will include the above clarifications in our revision to avoid such misunderstanding.
> > > >
> > > > ---
> > > >
> > > > **Q2:** The new convergence result you present also contains the IID case as an interesting special case. In that case , $\kappa=0$ and $\sigma=0$ in the deterministic case for each client. In that case however, it seems your result from Proposition 1 contradicts the 'Learning from history' lower bounds which show that algorithms which don't use history (such as yours with any AGR) can not possibly be Byzantine robust in the IID case?
> > > >
> > > > **A2:**
> > > > The lower bound from 'Learning from history' (Karimireddy et al., 2021) is posted below:
> > > >
> > > > **Theorem 2** (Failure of permutation-invariant methods). Suppose we are given any permutation invariant algorithm AGG as in Definition B, $\mu\ge0$, $\delta\in[0,1]$, and $n$ large enough that $\delta n\ge4(1+\log t)$. Then, there exists a $\delta$-robust $\mu$ strongly-convex optimization problem satisfying Definition A, such that the output $\tilde{x}_t$ of ALG after $t$ rounds necessarily has error
> > > > $$\mathbb{E}[f(\tilde{x}_t)]-f(x^*)\ge\Omega(\frac{\delta\sigma^2}{\mu}).$$
> > > >
> > > > In Theorem 2, $\sigma$ characterizes the randomness of SGD, which is consistent with our definition of $\sigma$. Therefore, in the mentioned special case ($\kappa=0$ and $\sigma=0$), $\sigma$ in Theorem 2 also becomes zero. Then the lower bound term $\Omega(\frac{\delta\sigma^2}{\mu})$ on the right-hand-side of the equation in Theorem 2 becomes zero.
> > > >
> > > > Thus, their lower bound is not contradictory to our upper bound in Proposition 1.

---

> > > > > ### Comment · Area_Chair_GocS · 2022-11-26
> > > > > **Assumptions need to be clear for main result**
> > > > >
> > > > > Thanks. A2: you're right about the lower bound, there is no issue there.
> > > > >
> > > > > However for the bigger question A1, the main proposition, you should make all assumptions clear in the main paper, and the proof. Could you clarify this in a new revision?

---

> > > > > > ### Author Response · Authors · 2022-11-27
> > > > > > **Author Response**
> > > > > >
> > > > > > Sorry that we are not allowed to revise our paper in the current stage. We will clarify these issues in our revision and upload it as soon as we are allowed.
> > > > > >
> > > > > > ---
> > > > > >
> > > > > > All assumptions for Proposition 1 are listed as follows.
> > > > > >
> > > > > > **Assumption 1** (Unbiased Estimator). The stochastic gradients sampled from any local data distribution are unbiased estimators of local gradients over $\mathbb{R}^{d}$ for all clients, i.e.,
> > > > > > $$\mathbb{E}_{\xi_i^t}[\nabla\mathcal{L}_i(w; \xi_i^t)]=\nabla\mathcal{L}_i(w),
> > > > > > \quad\forall w \in\mathbb{R}^{d}, i\in[n], t\in\mathbb{N}^+.$$
> > > > > >
> > > > > > **Assumption 2** (Bounded Variance). The variance of stochastic gradients sampled from any local data distribution is uniformly bounded over $\mathbb{R}^{d}$ for all clients, i.e., there exists $\sigma\ge0$ such that
> > > > > > $$\mathbb{E}\Vert\nabla\mathcal{L}_i(w; \xi_i^t)-\nabla\mathcal{L}_i(w)\Vert ^2\le\sigma^2,
> > > > > > \quad\forall w\in\mathbb{R}^{d}, i\in[n], t\in\mathbb{N}^+.$$
> > > > > >
> > > > > > **Assumption 3** (Gradient Dissimilarity). The difference between the local gradients and the global gradient is uniformly bounded over $\mathbb{R}^{d}$ for all clients, i.e., there exists $\kappa\ge0$ such that
> > > > > > $$\Vert\nabla\mathcal{L}_i(w)-\nabla\mathcal{L}(w)\Vert^2 \le\kappa^2,
> > > > > > \quad\forall w\in\mathbb{R}^{d}, i\in[n].$$
> > > > > >
> > > > > > **Assumption 4** (Lipschitz Smoothness). The loss function is $L$-Lipschitz smooth with respect over $\mathbb{R}^d$, i.e.,
> > > > > > $$\Vert\nabla\mathcal{L}(w)-\nabla\mathcal{L}(w')\Vert\le\Vert w-w'\Vert, \quad\forall w,w'\in \mathbb{R}^d.$$
> > > > > >
> > > > > > And we further assume the $c$ resilience of the base aggregator $\mathcal{A}$.
> > > > > >
> > > > > > **Definition 1** ($c$-resilient). Let $\mathcal{A}$ be an AGR. If for any input $\{x_1,...,x_n\}$ such that there exists a set $\mathcal{H}\in[n]$ of size at least $|\mathcal{H}|>n/2$ that satisfies:
> > > > > > $$\mathbb{E}\Vert x_i-x_{i'}\Vert^2\le\rho^2,\quad\forall i,i'\in\mathcal{H},$$
> > > > > > the output of $\mathcal{A}$ satisfies:
> > > > > > $$\mathbb{E}\Vert \mathcal{A} (x_1,...,x_n)-x\Vert^2\le c\rho^2,\text{where }x=\frac{1}{|\mathcal{H}|}\sum_{h\in\mathcal{H}}x_h,$$
> > > > > > Then the AGR $\mathcal{A}$ is called $c$-resilient
> > > > > >
> > > > > > Please refer to Page 6 of the paper for more details.
> > > > > >
> > > > > > ---
> > > > > >
> > > > > > We will also provide an overview of our proof for Proposition 1 in the main paper and explain more about the role of Lemma 1 in the revision.

---

> > ### Author Response · Authors · 2022-11-21
> > **Response to Q3**
> >
> > **Q3:** Could you clarify or give a reference how the earlier aggregation rules (page 6) are c-resilient AGR?
> >
> > **A3:**
> > Our definition of $c$-resilient AGR (Definition 1) is a randomized version of a popular definition $(f, \lambda)$-resilience (Definition 2) [2]. [2] shows that a number of AGRs (e.g., Blanchard et al. 2017, Pillutla et al., 2019, Yin et al., 2018) satisfy the $(f, \lambda)$-resilient definition. We show that an $(f, \lambda)$-resilient definition is $c$-resilient for $c=\lambda$ (Proposition 2). Thus, a number of AGRs (e.g., Blanchard et al. 2017, Pillutla et al., 2019, Yin et al., 2018) are $c$-resilient. We will discuss more details in our final version.
> >
> > **Definition 1 ($c$-resilient)** Let $A$ be an AGR. If for any input $\{x_1,...,x_n \}$ such that there exists a set $H \in[n]$ of size at least $|H|>n/2$ that satisfies:
> > $$\mathbb{E}\Vert x_i-x_{i'}\Vert ^2\le\rho^2,\quad\forall i,i'\in H,$$
> > the output of $A$ satisfies:
> > $$\mathbb{E}\Vert A (x_1,...,x_n) - x\Vert ^2\le c\rho^2, \text{where }x=\frac{1}{|H|}\sum_{h\in H}x_h, $$
> > then the AGR $A $ is called $c$-resilient.
> >
> > **Definition 2 ($(f, \lambda)$-resilient)** For $f<n$ and real value $\lambda\ge0$, an aggregation rule $A$ is called $(f, \lambda)$-resilient averaging if for any collection of $n$ vectors $x_1, ..., x_n$ and any set $S\subseteq\{1,...,n\}$ of size $n-f$,
> >
> > $$\Vert A(x_1,...,x_n)-\bar{x}_S\Vert\le\lambda\max _{i,j\in S}\Vert x_i-x_j\Vert $$
> >
> > where $\bar{x}_S:=\sum _{i\in S}x_i/|S|$, and $|S|$ is the cardinality of $S$.
> >
> > **Proposition 2** An $(f, \lambda)$-resilient aggregator is $c$-resilient for $c=\lambda$.
> >
> > **Proof:** Let $A$ denote the $(f, \lambda)$-resilient aggregator. For any input $\{x_1,...,x_n \}$, take $H \in[n]$, $|H|>n/2$ and $\rho\ge\mathbb{E}\Vert x_i-x_{i'}\Vert^2, \forall i,i'\in H $.
> > Since $A$ is $(f, \lambda)$-resilient,
> > $$\Vert A(x_1,...,x_n)-\bar{x}_S\Vert\le\lambda\max _{i,j\in S}\Vert x_i-x_j\Vert.$$
> > Take expectation on both sides of the above inequality, then we have
> > $$\mathbb{E}\Vert A(x_1,...,x_n)-\bar{x}_S\Vert\le\lambda\mathbb{E}\max _{i,j\in S}\Vert x_i-x_j\Vert \le\lambda\rho^2 $$
> >
> > Therefore, $A$ is $c$-resilient with $c=\lambda$.
> >
> > ---
> >
> > **References:**
> >
> > [2] Farhadkhani, Sadegh, et al. "Byzantine Machine Learning Made Easy by Resilient Averaging of Momentums." arXiv preprint arXiv:2205.12173 (2022)

---

> > ### Author Response · Authors · 2022-11-21
> > **Response to Q4**
> >
> > **Q4:** Could you comment on how your convergence theory compares to recent robust non-IID convergence results such as (Karimireddy et al. 2022), (El-Mhamdi et al. 2021), as well as this related additional reference [1]. Also the lower bounds of impossibility of learning in the non-IID case even in low-dimensions?
> >
> > **A4:**
> >
> > **Comparison:**
> > We all provided convergence guarantees of the same type, i.e., we all guarantee that we can reach an approximate optimal point after a certain number of communication rounds.
> >
> > We want to emphasize that all the convergence analysis comes from different aspects of defending against Byzantines. In particular, (Karimireddy et al. 2022) consider the impact of bucket together with momentum; (El-Mhamdi et al. 2021) focus on the aggregation rules; [1] considers a combined effect of client variance reduction and aggregation rules; and we concentrate more on gradient dimensions. From this perspective, our result is orthogonal to previous ones.
> >
> > More detailed differences are listed as follows: (El-Mhamdi et al. 2021) focus on decentralized FL with a server and provide an order optimal upper bound. However, this strong result requires a Byzantine ratio lower than $1/3$. By contrast, we consider a centralized FL setting and only assumes the Byzantine ratio to be lower than $1/2$. [1] consider an ideal case where the objective function is strongly convex, while we consider a more general non-convex case. We will discuss more details in our final version.
> >
> > **Lower bound:** (Karimireddy et al. 2022) have provided a lower bound for the general non-IID setting, which is applied to Byzantine-robust learning in both high-dimensional and low-dimensional cases. These results verify that even in low-dimensions, FL cannot be completely relieved from Byzantines due to the non-IID data. We will discuss more in our final version.
> >
> > ---
> >
> > **References:**
> >
> > [1] Byzantine-Robust Variance-Reduced Federated Learning over Distributed Non-i.i.d. Data Jie Peng, Zhaoxian Wu, Qing Ling, Tianyi Chen https://arxiv.org/abs/2009.08161

---

> > ### Author Response · Authors · 2022-11-21
> > **Response to Q5**
> >
> > **Q5:** Could you include such recent aggregators specifically for non-IID also in your experiments as baselines, such as (Karimireddy et al. 2022, El-Mhamdi et al. 2021), and [1]?
> >
> > **A5:**
> > We have already included aggregator RBTM (El-Mhamdi et al. 2021) in our paper. We run new experiments on bucketing (El-Mhamdi et al. 2021) and VR [1] and the results are posted in Table 1. As shown in Table 1, our GAIN outperforms Bucketing. Both GAIN and Bucketing improve the performance of VR, and the improvement effect of GAIN is greater. We will post more empirical results in our final revision.
> >
> > Table 1: Accuracy of different defenses under 6 attacks on CIFAR-10.
> >
> > | Attack | BitFlip | LabelFlip | LIE | Min-Max | Min-Sum | IPM |
> > |:-:|-:|-:|-:|-:|-:|-:|
> > | Multi-Krum+Bucketing | 47.87 | 49.86 | 45.90 | 43.53 | 44.92 | 50.28 |
> > | Multi-Krum+GAIN | **59.23** | **61.47** | **55.66** | **49.19** | **53.59** | **56.94** |
> > ||
> > | Bulyan+Bucketing | 51.79 | 61.16 | 46.02 | 45.90 | 52.30 | 56.44 |
> > | Bulyan+GAIN | **59.14** | **61.21** | **48.90** | **48.35** | **53.74** | **56.53** |
> > ||
> > | Median+Bucketing | 53.17 | 59.50 | 46.13 | 47.93 | 51.52 | 52.69 |
> > | Median+GAIN | **59.28** | **61.24** | **46.60** | **49.37** | **53.32** | **56.33** |
> > ||
> > | RFA+Bucketing | 52.55 | 58.44 | 48.71 | 47.51 | 52.29 | 55.19 |
> > | RFA+GAIN | **53.35** | **62.25** | **52.69** | **52.64** | **56.16** | **62.26** |
> > ||
> > | DnC+Bucketing | 57.79 | 59.39 | 57.53 | 55.09 | 53.83 | 54.01 |
> > | DnC+GAIN | **58.96** | **61.02** | **61.87** | **61.04** | **54.36** | **57.92** |
> > ||
> > | RBTM+Bucketing | 53.25 | 60.10 | 51.87 | 49.32 | 53.56 | 53.77 |
> > | RBTM+GAIN | **59.41** | **60.75** | **52.10** | **49.60** | **53.63** | **56.65** |
> > ||
> > | VR | 52.71 | 53.41 | 55.46 | 37.33 | 39.19 | 45.42 |
> > | VR+Bucketing | 52.99 | 60.77 | 57.32 | 51.28 | 51.09 | 53.80 |
> > | VR+GAIN | **53.55** | **61.04** | **61.02** | **51.59** | **53.60** | **53.84** |
> >
> > ---
> >
> > **References:**
> >
> > [1] Byzantine-Robust Variance-Reduced Federated Learning over Distributed Non-i.i.d. Data Jie Peng, Zhaoxian Wu, Qing Ling, Tianyi Chen https://arxiv.org/abs/2009.08161

---

### Decision · Program_Chairs · 2023-01-20

**Decision:**

Reject

**Justification For Why Not Higher Score:**

Main result is not supported by proof yet. Reviewers were not able to assess the new main result in time as authors changed it last minute unfortunately

**Justification For Why Not Lower Score:**

N/A

**Metareview: Summary, Strengths And Weaknesses:**

The paper studies Byzantine robust training in the important case of heterogeneous data on each client.
Unfortunately the main theoretical result in the submitted version of the paper did not hold true as it did not address the Byzantine setting. During the discussions, the authors have replaced this result with a different one (Proposition 1). The proof in the appendix nevertheless still relied (on page 14) on the non-realistic assumption that Byzantine gradients must be separated from regular ones. While we have discussed in details with authors and reviewers, there was unfortunately not enough time to converge on an assessment of the new proof with the reviewers. A careful assessment of the new main results would require a new submission. Note that the negative review by Reviewer Bimn was discarded as low confidence.

We hope the detailed feedback helps to strengthen the paper for a future occasion.

**Summary Of Ac-Reviewer Meeting:**

We arranged to meet on zoom but reviewers had to cancel the meeting on short notice. We continued open-review discussions, where reviewers were not able to assess the new main result unfortunately. I've checked both versions of the main results and both its proof variants as it evolved